# ON THE LOSS LANDSCAPE OF A CLASS OF DEEP NEURAL NETWORKS WITH NO BAD LOCAL VALLEYS

**Quynh Nguyen**
Saarland University, Germany

**Mahesh Chandra Mukkamala**
Saarland University, Germany

**Matthias Hein**
University of Tübingen, Germany

## ABSTRACT

We identify a class of over-parameterized deep neural networks with standard activation functions and cross-entropy loss which provably have no bad local valley, in the sense that from any point in parameter space there exists a continuous path on which the cross-entropy loss is non-increasing and gets arbitrarily close to zero. This implies that these networks have no sub-optimal strict local minima.

## 1 INTRODUCTION

It has been empirically observed in deep learning (Dauphin et al., 2014; Goodfellow et al., 2015) that the training problem of over-parameterized[1] deep CNNs (LeCun et al., 1990; Krizhevsky et al., 2012) does not seem to have a problem with bad local minima. In many cases, local search algorithms like stochastic gradient descent (SGD) frequently converge to a solution with zero training error even though the training objective is known to be non-convex and potentially has many distinct local minima (Auer et al., 1996; Safran & Shamir, 2018). This indicates that the problem of training practical over-parameterized neural networks is still far from the worst-case scenario where the problem is known to be NP-hard (Blum & Rivest., 1989; Sima, 2002; Livni et al., 2014; Shalev-Shwartz et al., 2017). A possible hypothesis is that the loss landscape of these networks is "well-behaved" so that it becomes amenable to local search algorithms like SGD and its variants. As not all neural networks have a well-behaved loss landscape, it is interesting to identify sufficient conditions on their architecture so that this is guaranteed. In this paper our motivation is to come up with such a class of networks in a practically relevant setting, that is we study multi-class problems with the usual empirical cross-entropy loss and deep (convolutional) networks and almost no assumptions on the training data, in particular no distributional assumptions. Thus our results directly apply to the networks which we use in the experiments.

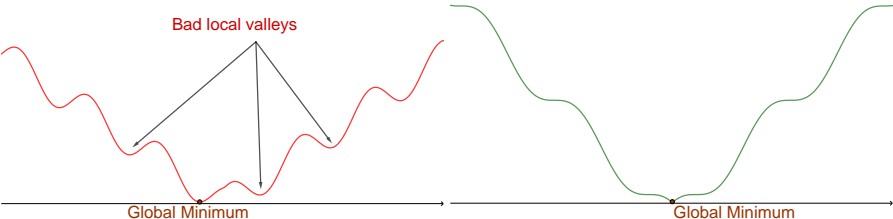

Figure 1: An example loss landscape with bad local valleys (left) and without bad local valley (right).

**Our contributions.** We identify a family of deep networks with skip connections to the output layer whose loss landscape has no bad local valleys (see Figure 1 for an illustration). Our setting is

---

[1]These are the networks which have more parameters than necessary to fit the training data

for the empirical loss and there are no distributional assumptions on the training data. Moreover, we study directly the standard cross-entropy loss for multi-class problems. There are little assumptions on the network structure which can be arbitrarily deep and can have convolutional layers (weight sharing) and skip-connections between hidden layers. From a practical perspective, one can generate an architecture which fulfills our conditions by taking an existing CNN architecture and then adding skip-connections from a random subset of $N$ neurons ($N$ is the number of training samples), possibly from multiple hidden layers, to the output layer (see Figure 2 for an illustration). For these networks we show that there always exists a continuous path from any point in parameter space on which the loss is non-increasing and gets arbitrarily close to zero. We note that this implies the loss landscape has no strict local minima, but theoretically non-strict local minima can still exist. Beside that, we show that the loss has also no local maxima.

Beside the theoretical analysis, we show in experiments that despite achieving zero training error, the aforementioned class of neural networks generalize well in practice when trained with SGD whereas an alternative training procedure guaranteed to achieve zero training error has significantly worse generalization performance and is overfitting. Thus we think that the presented class of neural networks offer an interesting test bed for future work to study the implicit bias/regularization of SGD.

## 2 DESCRIPTION OF NETWORK ARCHITECTURE

We consider a family of deep neural networks which have $d$ input units, $H$ hidden units, $m$ output units and satisfy the following conditions:

1. Every hidden unit of the first layer can be connected to an arbitrary subset of input units.

2. Every hidden unit at higher layers can take as input an arbitrary subset of hidden units from (multiple) lower hidden layers.

3. Any subgroup of hidden units lying on the same layer can have non-shared or shared weights, in the later case their number of incoming units have to be equal.

4. There exist $N$ hidden units which are connected to the output nodes with independent weights ($N$ denotes the number of training samples).

5. The output of every hidden unit $j$ in the network, denoted as $f_j : \mathbb{R}^d \to \mathbb{R}$, is given as

$$f_j(x) = \sigma_j \Big( b_j + \sum_{k : k \to j} f_k(x) u_{k \to j} \Big)$$

where $x \in \mathbb{R}^d$ is an input vector of the network, $\sigma_j : \mathbb{R} \to \mathbb{R}$ is the activation function of unit $j$, $b_j \in \mathbb{R}$ is the bias of unit $j$, and $u_{k \to j} \in \mathbb{R}$ the weight from unit $k$ to unit $j$.

This definition covers a class of deep fully connected and convolutional neural networks with an additional condition on the number of connections to the output layer. In particular, while conventional architectures have just connections from the last hidden layer to the output, we require in our setting that there must exist at least $N$ neurons, "regardless" of their hidden layer, that are connected to the output layer. Essentially, this means that if the last hidden layer of a traditional network has just $L < N$ neurons then one can add connections from $N - L$ neurons in the hidden layers below it to the output layer so that the network fulfills our conditions.

Similar skip-connections have been used in DenseNet (Huang et al., 2017) which are different from identity skip-connections as used in ResNets (He et al., 2016). In Figure 2 we illustrate a network with and without skip connections to the output layer which is analyzed in this paper. We note that several architectures like DenseNets Huang et al. (2017) already have skip-connections between hidden layers in their original architecture, whereas our special skip-connections go from hidden layers directly to the output layer. As our framework allow both kinds to exist in the same network (see Figure 2 for an example), we would like to separate them from each other by making the convention that in the following skip-connections, if not stated otherwise, always refer to ones which connect hidden neurons to output neurons.

We denote by $d$ the dimension of the input and index all neurons in the network from the input layer to the output layer as $1, 2, \ldots, d, d + 1, \ldots, d + H, d + H + 1, \ldots, d + H + m$ which correspond to $d$ input units, $H$ hidden units and $m$ output units respectively. As we only allow directed

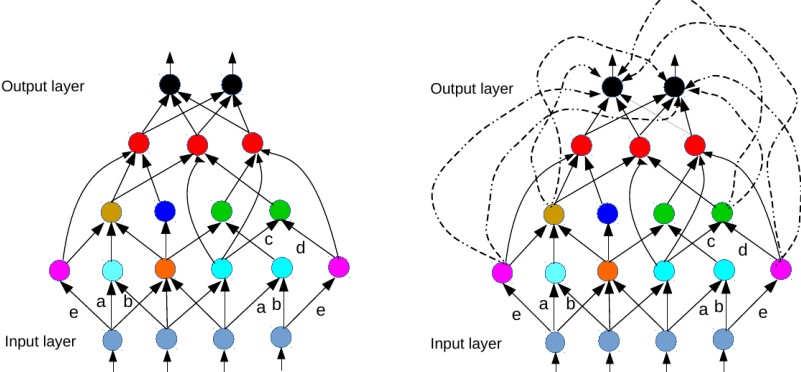

Figure 2: **Left**: An example neural network represented as directed acyclic graph. **Right**: The same network with skip connections added from a subset of hidden neurons to the output layer. All neurons with the same color can have shared or non-shared weights.

arcs from lower layers to upper layers, it follows that $k < j$ for every $k \rightarrow j$. Let $N$ be the number of training samples. Suppose that there are $M$ hidden neurons which are directly connected to the output with independent weights where it holds $N \leq M \leq H$. Let $\{p_1, \ldots, p_M\}$ with $p_j \in \{d+1, \ldots, d+H\}$ be the set of hidden units which are directly connected to the output units. Let $\text{in}(j)$ be the set of incoming nodes to unit $j$ and $u_j = [u_{k \rightarrow j}]_{k \in \text{in}(j)}$ the weight vector of the $j$-th unit. Let $U = (u_{d+1}, \ldots, u_{d+H}, b_{d+1}, \ldots, b_{d+H})$ denote the set of all weights and biases of all hidden units in the network. Let $V \in \mathbb{R}^{M \times m}$ be the weight matrix which connects the $M$ hidden neurons to the $m$ output units of the network. An important quantity in the following is the matrix $\Psi \in \mathbb{R}^{N \times M}$ defined as

$$\Psi = \begin{bmatrix} f_{p_1}(x_1) & \ldots & f_{p_M}(x_1) \\ \vdots & & \vdots \\ f_{p_1}(x_N) & \ldots & f_{p_M}(x_N) \end{bmatrix} \tag{1}$$

As $\Psi$ depends on $U$, we write $\Psi_U$ or $\Psi(U)$ as a function of $U$. Let $G \in \mathbb{R}^{N \times m}$ be the output of the network for all training samples. In particular, $G_{ij}$ is the value of the $j$-th output neuron for training sample $x_i$. It follows from our definition that

$$G_{ij} = \langle \Psi_{i:}, V_{:j} \rangle = \sum_{k=1}^{M} f_{p_k}(x_i) V_{kj}, \quad \forall i \in [N], j \in [m]$$

Let $(x_i, y_i)_{i=1}^{N}$ be the training set where $y_i$ denotes the target class for sample $x_i$. In the following we analyze the commonly used cross-entropy loss given as

$$\Phi(U, V) = \frac{1}{N} \sum_{i=1}^{N} -\log \left( \frac{e^{G_{iy_i}}}{\sum_{k=1}^{m} e^{G_{ik}}} \right) \tag{2}$$

We refer to Section C in the appendix for extension of our results to general convex losses. The cross-entropy loss is bounded from below by zero but this value is not attained. In fact the global minimum of the cross-entropy loss need not exist e.g. if a classifier achieves zero training error then by upscaling the function to infinity one can drive the loss arbitrarily close to zero. Due to this property, we do not study the global minima of the cross-entropy loss but the question if and how one can achieve zero training error. Moreover, we note that sufficiently small cross-entropy loss implies zero training error as shown in the following lemma.

**Lemma 2.1** *If $\Phi(U, V) < \frac{\log(2)}{N}$, then the training error is zero.*

**Proof:** We note that if $\Phi(U, V) < \frac{\log(2)}{N}$, then it holds due to the positivity of the loss,

$$\max_{i=1, \ldots, N} -\log \left( \frac{e^{G_{iy_i}}}{\sum_{k=1}^{m} e^{G_{ik}}} \right) \leq \sum_{i=1}^{N} -\log \left( \frac{e^{G_{iy_i}}}{\sum_{k=1}^{m} e^{G_{ik}}} \right) < \log(2).$$

This implies that for all $i = 1, \ldots, N$,

$$\log \big( 1 + \sum_{k \neq y_i} e^{G_{ik} - G_{iy_i}} \big) < \log(2) \quad \implies \quad \sum_{k \neq y_i} e^{G_{ik} - G_{iy_i}} < 1.$$

In particular: $\max_{k \neq y_i} e^{G_{ik} - G_{iy_i}} < 1$ and thus $\max_{k \neq y_i} G_{ik} - G_{iy_i} < 0$ for all $i = 1, \ldots, N$ which implies the result. $\qquad\square$

## 3 MAIN RESULT

The following conditions are required for the main result to hold.

**Assumption 3.1**    *1. All activation functions $\{\sigma_{d+1}, \ldots, \sigma_{d+H}\}$ are real analytic and strictly increasing*

2. *Among $M$ neurons $\{p_1, \ldots, p_M\}$ which are connected to the output units, there exist $N \leq M$ neurons, say w.l.o.g. $\{p_1, \ldots, p_N\}$, such that one of the following conditions hold:*

   - *For every $1 \leq j \leq N$: $\sigma_{p_j}$ is bounded and $\lim_{t \to -\infty} \sigma_{p_j}(t) = 0$*
   - *For every $1 \leq j \leq N$: $\sigma_{p_j}$ is the softplus activation (3), and there exists a backward path from $p_j$ to the first hidden layer s.t. on this path there is no neuron which has skip-connections to the output or shared weights with other skip-connection neurons.*

3. *The input patches of different training samples are distinct. In particular, let $n_1$ be the number of units in the first hidden layer and denote by $S_i$ for $i \in [d+1, d+n_1]$ their input support, then for all $r \neq s \in [N]$, and $i \in [d+1, d+n_1]$, it holds $x_r|_{S_i} \neq x_s|_{S_i}$.*

The first condition of Assumption 3.1 is satisfied for softplus, sigmoid, tanh, etc, whereas the second condition is fulfilled for sigmoid and softplus. For softplus activation function (smooth approximation of ReLU),

$$\sigma_\gamma(t) = \frac{1}{\gamma} \log(1 + e^{\gamma t}), \quad \text{for some } \gamma > 0, \tag{3}$$

we require an additional assumption on the network architecture. The third condition is always satisfied for fully connected networks if the training samples are distinct. For CNNs, this condition means that the corresponding input patches across different training samples are distinct. This could be potentially violated if the first convolutional layer has very small receptive fields. However, if this condition is violated for the given training set then after an arbitrarily small random perturbation of all training inputs it will be satisfied with probability 1. Note that the $M$ neurons which are directly connected to the output units can lie on different hidden layers in the network. Also there is no condition on the width of every individual hidden layer as long as the total number of hidden neurons in the network is larger than $N$ so that our condition $M \geq N$ is feasible.

Overall, we would like to stress that Assumption 3.1 covers a quite large class of interesting network architectures but nevertheless allows us to show quite strong results on their empirical loss landscape.

The following key lemma shows that for almost all $U$, the matrix $\Psi(U)$ has full rank.

**Lemma 3.2** *Under Assumption 3.1, the set of $U$ such that $\Psi(U)$ has not full rank $N$ has Lebesgue measure zero.*

**Proof:** (Proof sketch) Due to space limitation, we can only present below a proof sketch. We refer the reader to the appendix for the detailed proof. The proof consists of two main steps. First, we show that there exists $U$ s.t. the submatrix $\Psi_{1:N,1:N}$ has full rank. By Assumption 3.1, all activation functions are real analytic, thus the determinant of $\Psi_{1:N,1:N}$ is a real analytic function of the network parameters which $\Psi$ depends on. By the first result, this determinant function is not identically zero, thus Lemma A.1 shows that the set of $U$ for which $\Psi$ has not full rank has Lebesgue measure zero.

Now we sketch the proof for the first step, that is to find a $U$ s.t. $\Psi$ has full rank. By Assumption 3.1.3, one can always choose the weight vectors of the first hidden layer so that every neuron at this

layer has distinct values for different training samples. For higher neurons, we set their initial weight vectors to be unit vectors with exactly one $1$ and $0$ elsewhere. Note that the above construction of weights can be easily done so that all the neurons from the same layer and with the same number of incoming units can have shared/unshared weights according to our description of network architecture in Section 2. Let $c(j)$ be the neuron below $j$ s.t. $u_{c(j)\to j} = 1$. To find $U$, we are going to scale up each weight vector $u_j$ by a positive scalar $\alpha_j$. The idea is to show that the determinant of $\Psi_{1:N,1:N}$ is non-zero for some positive value of $\{\alpha_j\}$. The biases can be chosen in such a way that the following holds for some $\beta \in \mathbb{R}$,

$$\Psi_{ij} = f_{p_j}(x_i) = \sigma_{p_j}\Big(\beta + \alpha_{p_j}\big(f_{c(p_j)}(x_i) - f_{c(p_j)}(x_j)\big)\Big) \quad \forall j \in [N] \tag{4}$$

where $f_{c(p_j)}(x) = \sigma_{c(p_j)}(\alpha_{c(p_j)}\sigma_{c(c(p_j))}(\ldots f_{q_j}(x)\ldots))$ with $q_j$ being the index of some neuron in the first hidden layer. Note that by our construction the value of unit $q_j$ is distinct at different training samples, and thus it follows from the strict monotonic property of activation functions from Assumption 3.1 and the positivity of $\{\alpha_{c(p_j)},\ldots\}$ that $f_{c(p_j)}(x_i) \neq f_{c(p_j)}(x_j)$ for every $i \neq j$. Next, we show that the set of training samples can be re-ordered in such a way that it holds $f_{c(p_j)}(x_i) < f_{c(p_j)}(x_j)$ for every $i > j$. Note that this re-ordering does not affect the rank of $\Psi$. Now, the intuition is that if one let $\alpha_{p_j}$ go to infinity then $\Psi_{ij}$ converges to zero for $i > j$ because it holds for all activations from Assumption 3.1 that $\lim_{t\to-\infty} \sigma_{p_j}(t) = 0$. Thus the determinant of $\Psi_{1:1,1:N}$ converges to $\prod_{i=1}^{N} \sigma_{p_j}(\beta)$ which can be chosen to be non-zero by a predefined value of $\beta$, in which case $\Psi$ will have full rank. The detailed proof basically will show how to choose the specific values of $\{\alpha_j\}$ so that all the above criteria are met. In particular, it is important to make sure that the weight vectors of two neurons $j, j'$ from the same layer will be scaled by the same factor $\alpha_j = \alpha_{j'}$ as we want to maintain any potential weight-sharing conditions. The choice of activation functions from the second condition of Assumption 3.1 basically determines how the values of $\alpha$ should be chosen. $\square$

While we conjecture that the result of Lemma 3.2 holds for softplus activation function without the additional condition as mentioned in Assumption 3.1, the proof of this is considerably harder for such a general class of neural networks since one has to control the output of neurons with skip connection from different layers which depend on each other. However, please note that the condition is also not too restrictive as it just might require more connections from lower layers to upper layers but it does not require that the network is wide. Before presenting our main result, we first need a formal definition of bad local valleys.

**Definition 3.3** *The $\alpha$-sublevel set of $\Phi$ is defined as $L_\alpha = \{(U, V) \mid \Phi(U, V) < \alpha\}$. A local valley is defined as a connected component of some sublevel set $L_\alpha$. A bad local valley is a local valley on which the loss function $\Phi$ cannot be made "arbitrarily small".*

Intuitively, a typical example of a bad local valley is a small neighborhood around a sub-optimal strict local minimum. We are now ready to state our main result.

**Theorem 3.4** *The following holds under Assumption 3.1:*

1. *There exist uncountably many solutions with zero training error.*

2. *The loss landscape of $\Phi$ does not have any bad local valley.*

3. *There exists no suboptimal strict local minimum.*

4. *There exists no local maximum.*

**Proof:**

1. By Lemma 3.2 the set of $U$ such that $\Psi(U)$ has not full rank $N$ has Lebesgue measure zero. Given $U$ such that $\Psi$ has full rank, the linear system $\Psi(U)V = Y$ has for every possible target output matrix $Y \in \mathbb{R}^{N\times m}$ at least one solution $V$. As this is possible for almost all $U$, there exist uncountably many solutions achieving zero training error.

2. Let $C$ be a non-empty, connected component of some $\alpha$-sublevel set $L_\alpha$ for $\alpha > 0$. Suppose by contradiction that the loss on $C$ cannot be made arbitrarily small, that is there exists

an $\epsilon > 0$ such that $\Phi(U, V) \geq \epsilon$ for all $(U, V) \in C$, where $\epsilon < \alpha$. By definition, $L_\alpha$ can be written as the pre-image of an open set under a continuous function, that is $L_\alpha = \Phi^{-1}(\{a \mid a < \alpha\})$, and thus $L_\alpha$ must be an open set (see Proposition A.2). Since $C$ is a non-empty connected component of $L_\alpha$, $C$ must be an open set as well, and thus $C$ has non-zero Lebesgue measure. By Lemma 3.2 the set of $U$ where $\Psi(U)$ has not full rank has measure zero and thus $C$ must contain a point $(U, V)$ such that $\Psi(U)$ has full rank. Let $Y$ be the usual zero-one one-hot encoding of the target network output. As $\Psi(U)$ has full rank, there always exist $V^*$ such that $\Psi(U)V^* = Yt^*$, where $t^* = \log\left(\frac{m-1}{e^{\frac{\epsilon}{2}}-1}\right)$ Note that the loss of $(U, V^*)$ is

$$\Phi(U, V^*) = -\log\left(\frac{e^{t^*}}{e^{t^*} + (m-1)}\right) = \log(1 + (m-1)e^{-t^*}) = \frac{\epsilon}{2}.$$

As the cross-entropy loss $\Phi(U, V)$ is convex in $V$ and $\Phi(U, V) < \alpha$ we have for the line segment $V(\lambda) = \lambda V + (1 - \lambda)V^*$ for $\lambda \in [0, 1]$,

$$\Phi(U, V(\lambda)) \leq \lambda\Phi(U, V) + (1 - \lambda)\Phi(U, V^*) < \lambda\alpha + (1 - \lambda)\frac{\epsilon}{2} < \alpha.$$

Thus the whole line segment is contained in $L_\alpha$ and as $C$ is a connected component it has to be contained in $C$. However, this contradicts the assumption that for all $(U, V) \in C$ it holds $\Phi(U, V) \geq \epsilon$. Thus on every connected component $C$ of $L_\alpha$ the training loss can be made arbitrarily close to zero and thus the loss landscape has no bad valleys.

3. Let $(U_0, V_0)$ be a strict suboptimal local minimum, then there exists $r > 0$ such that $\Phi(U, V) > \Phi(U_0, V_0) > 0$ for all $(U, V) \in B((U_0, V_0), r) \setminus \{(U_0, V_0)\}$ where $B(\cdot, r)$ denotes a closed ball of radius $r$. Let $\alpha = \min_{(U,V) \in \partial B\left((U_0, V_0), r\right)} \Phi(U, V)$ which exists as $\Phi$ is continuous and the boundary $\partial B\left((U_0, V_0), r\right)$ of $B\left((U_0, V_0), r\right)$ is compact. Note that $\alpha > \Phi(U_0, V_0)$ as $(U_0, V_0)$ is a strict local minimum. Consider the sub-level set $D = L_{\frac{\alpha+\Phi(U_0,V_0)}{2}}$. As $\Phi(U_0, V_0) < \frac{\alpha+\Phi(U_0,V_0)}{2}$ it holds $(U_0, V_0) \in D$. Let $E$ be the connected component of $D$ which contains $(U_0, V_0)$, that is, $(U_0, V_0) \in E \subseteq D$. It holds $E \subset B\left((U_0, V_0), r\right)$ as $\Phi(U, V) < \frac{\alpha+\Phi(U_0,V_0)}{2} < \alpha$ for all $(U, V) \in E$. Moreover, $\Phi(U, V) \geq \Phi(U_0, V_0) > 0$ for all $(U, V) \in E$ and thus $\Phi$ can not be made arbitrarily small on a connected component of a sublevel set of $\Phi$ and thus $E$ would be a bad local valley which contradicts 3.3.2.

4. Suppose by contradiction that $(U, V)$ is a local maximum. Then the Hessian of $\Phi$ is negative semi-definite. However, as principal submatrices of negative semi-definite matrices are again negative semi-definite, then also the Hessian of $\Phi$ w.r.t $V$ must be negative semi-definite. However, $\Phi$ is always convex in $V$ and thus its Hessian restricted to $V$ is positive semi-definite. The only matrix which is both p.s.d. and n.s.d. is the zero matrix. It follows that $\nabla_V^2 \Phi(U, V) = 0$. One can easily show that

$$\nabla_{V_{:j}}^2 \Phi = \sum_{i=1}^{N} \frac{e^{G_{ij}}}{\sum_{k=1}^m e^{G_{ik}}}\left(1 - \frac{e^{G_{ij}}}{\sum_{k=1}^m e^{G_{ik}}}\right)\Psi_{i:}\Psi_{i:}^T$$

From Assumption 3.1 it holds that there exists $j \in [N]$ s.t. $\sigma_{p_j}$ is strictly positive, and thus some entries of $\Psi_{i:}$ must be strictly positive. Moreover, one has $\frac{e^{G_{ij}}}{\sum_{k=1}^m e^{G_{ik}}} \in (0, 1)$. It follows that some entries of $\nabla_{V_{:j}}^2 \Phi$ must be strictly positive. Thus $\nabla_{V_{:j}}^2 \Phi$ cannot be identically zero, leading to a contradiction. Therefore $\Phi$ has no local maximum.

$\square$

Theorem 3.4 shows that there are infinitely many solutions which achieve zero training error, and the loss landscape is nice in the sense that from any point in the parameter space there exists a continuous path that drives the loss arbitrarily close to zero (and thus a solution with zero training error) on which the loss is non-increasing.

While the networks are over-parameterized, we show in the next Section 4 that the modification of standard networks so that they fulfill our conditions leads nevertheless to good generalization

performance, often even better than the original network. We would like to note that the proof of Theorem 3.4 also suggests a different algorithm to achieve zero training error: one initializes all weights, except the weights to the output layer, randomly (e.g. Gaussian weights), denoted as $U$, and then just solves the linear system $\Psi(U)V = Y$ to obtain the weights $V$ to the output layer. Basically, this algorithm uses the network as a random feature generator and fits the last layer directly to achieve zero training error. The algorithm is successful with probability 1 due to Lemma 3.2. Note that from a solution with zero training error one can drive the cross-entropy loss to zero by upscaling to infinity but this does not change the classifier. We will see, that this simple algorithm shows bad generalization performance and overfitting, whereas training the full network with SGD leads to good generalization performance. This might seem counter-intuitive as our networks have more parameters than the original networks but is inline with recent observations in Zhang et al. (2017) that state-of-the art networks, also heavily over-parameterized, can fit even random labels but still generalize well on the original problem. Due to this qualitative difference of SGD and the simple algorithm which both are able to find solutions with zero training error, we think that our class of networks is an ideal test bed to study the implicit regularization/bias of SGD, see e.g. Soudry et al. (2018).

## 4 EXPERIMENTS

The main purpose of this section is to investigate the generalization ability of practical neural networks with skip-connections added to the output layer to fulfill Assumption 3.1.

**Datasets.** We consider MNIST and CIFAR10 datasets. MNIST contains $5.5 \times 10^4$ training samples and $10^4$ test samples, and CIFAR10 has $5 \times 10^4$ training samples and $10^4$ test samples. We do not use any data pre-processing nor data-augmentation in all of our experiments.

**Network architectures.** For MNIST, we use a plain CNN architecture with 13 layers, denoted as CNN13 (see Table 3 in the appendix for more details about this architecture). For CIFAR10 we use VGG11, VGG13, VGG16 (Simonyan & Zisserman, 2015) and DenseNet121 (Huang et al., 2017). As the VGG models were originally proposed for ImageNet and have very large fully connected layers, we adapted these layers for CIFAR10 by reducing their width from 4096 to 128. For each given network, we create the corresponding skip-networks by adding skip-connections to the output so that our condition $M \geq N$ from the main theorem is satisfied. In particular, we aggregate all neurons of all the hidden layers in a pool and randomly choose from there a subset of $N$ neurons to be connected to the output layer (see e.g. Figure 2 for an illustration). As existing network architectures have a large number of feature maps per layer, the total number of neurons is often very large compared to number of training samples, thus it is easy to choose from there a subset of $N$ neurons to connect to the output. In the following, we test both sigmoid and softplus activation function ($\gamma = 20$) for each network architecture and their skip-variants. We use the standard cross-entropy loss and train all models with SGD+Nesterov momentum for 300 epochs. The initial learning rate is set to 0.1 for Densenet121 and 0.01 for the other architectures. Following Huang et al. (2017), we also divide the learning rate by 10 after 50% and 75% of the total number of training epochs. Note that we do not use any explicit regularization like weight decay or dropout.

The main goal of our experiments is to investigate the influence of the additional skip-connections to the output layer on the generalization performance. We report the test accuracy for the original models and the ones with skip-connections to the output layer. For the latter one we have two different algorithms: standard SGD for training the full network as described above (SGD) and the randomized procedure (rand). The latter one uses a slight variant of the simple algorithm described at the end of the last section: randomly initialize the weights of the network $U$ up to the output layer by drawing each of them from a truncated Gaussian distribution with zero mean and variance $\frac{2}{d}$ where $d$ is the number of weight parameters and the truncation is done after $\pm 2$ standard deviations (standard keras initialization), then use SGD to optimize the weights $V$ for a linear classifier with fixed features $\Psi(U)$ which is a convex optimization problem.

Our experimental results are summarized in Table 1 for MNIST and Table 2 for CIFAR10. For skip-models, we report mean and standard deviation over 8 random choices of the subset of $N$ neurons connected to the output.

**Discussion of results.** First of all, we note that adding skip connections to the output improves the test accuracy in almost all networks (with the exception of Densenet121) when the full network is

| | Sigmoid activation function | Softplus activation function |
|---|---|---|
| CNN13 | 11.35 | 99.20 |
| CNN13-skip (SGD) | $98.40 \pm 0.07$ | $99.14 \pm 0.04$ |

Table 1: Test accuracy (%) of CNN13 on MNIST dataset. CNN13 denotes the original architecture from Table 3 while CNN13-skip denotes the corresponding skip-model. There are in total $179, 840$ hidden neurons from the original CNN13 (see Table 3), out of which we choose a random subset of $N = 55, 000$ neurons to connect to the output layer to obtain CNN13-skip.

| | Sigmoid activation function | | Softplus activation function | |
|---|---|---|---|---|
| Model | Test acc (%) | Train acc (%) | Test acc (%) | Train acc (%) |
| VGG11 | 10 | 10 | 78.92 | 100 |
| VGG11-skip (rand) | $62.81 \pm 0.39$ | 100 | $64.49 \pm 0.38$ | 100 |
| VGG11-skip (SGD) | $\mathbf{72.51} \pm 0.35$ | 100 | $\mathbf{80.57} \pm 0.40$ | 100 |
| VGG13 | 10 | 10 | 80.84 | 100 |
| VGG13-skip (rand) | $61.50 \pm 0.34$ | 100 | $61.42 \pm 0.40$ | 100 |
| VGG13-skip (SGD) | $\mathbf{70.24} \pm 0.39$ | 100 | $\mathbf{81.94} \pm 0.40$ | 100 |
| VGG16 | 10 | 10 | 81.33 | 100 |
| VGG16-skip (rand) | $61.57 \pm 0.41$ | 100 | $61.46 \pm 0.34$ | 100 |
| VGG16-skip (SGD) | $\mathbf{70.61} \pm 0.36$ | 100 | $\mathbf{81.91} \pm 0.24$ | 100 |
| Densenet121 | $\mathbf{86.41}$ | 100 | $\mathbf{89.31}$ | 100 |
| Densenet121-skip (rand) | $52.07 \pm 0.48$ | 100 | $55.39 \pm 0.48$ | 100 |
| Densenet121-skip (SGD) | $81.47 \pm 1.03$ | 100 | $86.76 \pm 0.49$ | 100 |

Table 2: Traning and test accuracy of several CNN architectures with/without skip-connections on CIFAR10 (no data-augmentation). For each original model A, A-skip denotes the corresponding skip-model in which a subset of $N$ hidden neurons "randomly selected" from the hidden layers are connected to the output units. For Densenet121, these neurons are randomly chosen from the first dense block. The names in open brackets (rand/SGD) specify how the networks are trained: **rand** ($U$ is randomized and fixed while $V$ is learned with SGD), **SGD** (both $U$ and $V$ are optimized with SGD). Additional experimental results with data-augmentation are shown in Table 5 in the appendix.

trained with SGD. In particular, for the sigmoid activation function the skip connections allow for all models except Densenet121 to get reasonable performance whereas training the original model fails. This effect can be directly related to our result of Theorem 3.4 that the loss landscape of skip-networks has no bad local valley and thus it is not difficult to reach a solution with zero training error (see Section F in the appendix for more detailed discussions on this issue, as well as Section E for a visual example which shows why the skip-models can succeed while the original models fail). The exception is Densenet121 which gets already good performance for the sigmoid activation function for the original model. We think that the reason is that the original Densenet121 architecture has already quite a lot of skip-connections between the hidden layers which thus improves the loss surface already so that the additional connections added to the output units are not necessary anymore.

The second interesting observation is that we do not see any sign of overfitting for the SGD version even though we have increased for all models the number of parameters by adding skip connections to the output layer and we know from Theorem 3.4 that for all the skip-models one can easily achieve zero training error. This is in line with the recent observation of Zhang et al. (2017) that modern heavily over-parameterized networks can fit everything (random labels, random input) but nevertheless generalize well on the original training data when trained with SGD. This is currently an active research area to show that SGD has some implicit bias (Neyshabur et al., 2017; Brutzkus et al., 2018; Soudry et al., 2018) which leads to a kind of regularization effect similar to the linear least squares problem where SGD converges to the minimum norm solution. Our results confirm that there is an implicit bias as we see a strong contrast to the (skip-rand) results obtained by using the network

as a random feature generator and just fitting the connections to the output units (*i.e. V*) which also leads to solutions with zero training error with probability 1 as shown in Lemma 3.2 and the proof of Theorem 3.4. For this version we see that the test accuracy gets worse as one is moving from simpler networks (VGG11) to more complex ones (VGG16 and Densenet121) which is a sign of overfitting. Thus we think that our class of networks is also an interesting test bed to understand the implicit regularization effect of SGD. It seems that SGD selects from the infinite pool of solutions with zero training error one which generalizes well, whereas the randomized feature generator selects one with much worse generalization performance.

## 5 RELATED WORK

In the literature, many interesting theoretical results have been developed on the loss surface of neural networks Yu & Chen (1995); Haeffele & Vidal (2017); Choromanska et al. (2015); Kawaguchi (2016); Safran & Shamir (2016); Hardt & Ma (2017); Yun et al. (2017); Lu & Kawaguchi (2017); Venturi et al. (2018); Liang et al. (2018b); Zhang et al. (2018); Nouiehed & Razaviyayn (2018). The behavior of SGD for the minimization of training objective has been also analyzed for various settings (Andoni et al., 2014; Sedghi & Anandkumar, 2015; Janzamin et al., 2016; Gautier et al., 2016; Brutzkus & Globerson, 2017; Soltanolkotabi, 2017; Soudry & Hoffer, 2017; Zhong et al., 2017; Tian, 2017; Du et al., 2018; Wang et al., 2018) to name a few. Most of current results are however limited to shallow networks (one hidden layer), deep linear networks and/or making simplifying assumptions on the architecture or the distribution of training data. An interesting recent exception is Liang et al. (2018a) where they show that for binary classification one neuron with a skip-connection to the output layer and exponential activation function is enough to eliminate all bad local minima under mild conditions on the loss function. More closely related in terms of the setting are (Nguyen & Hein, 2017; 2018) where they study the loss surface of fully connected and convolutional networks if one of the layers has more neurons than the number of training samples for the standard multi-class problem. However, the presented results are stronger as we show that our networks do not have any suboptimal local valley or strict local minima and there is less over-parameterization if the number of classes is small.

## 6 CONCLUSION

We have identified a class of deep neural networks whose loss landscape has no bad local valleys. While our networks are over-parameterized and can easily achieve zero training error, they generalize well in practice when trained with SGD. Interestingly, a simple different algorithm using the network as random feature generator also achieves zero training error but has significantly worse generalization performance. Thus we think that our class of models is an interesting test bed for studying the implicit regularization effect of SGD.

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

## A  MATHEMATICAL TOOLS

In the proof of Lemma 3.2 we make use of the following property of analytic functions.

**Lemma A.1** *(Nguyen, 2015; Mityagin, 2015) If $f : \mathbb{R}^n \to \mathbb{R}$ is a real analytic function which is not identically zero then the set $\{x \in \mathbb{R}^n \mid f(x) = 0\}$ has Lebesgue measure zero.*

We recall the following standard result from topology (see *e.g.* Apostol (1974), Theorem 4.23, p. 82), which is used in the proof of Theorem 3.4.

**Proposition A.2** *Let $f : \mathbb{R}^m \to \mathbb{R}^n$ be a continuous function. If $U \subseteq \mathbb{R}^n$ is an open set then $f^{-1}(U)$ is also open.*

## B  PROOF OF LEMMA 3.2

**Proof:** We assume w.l.o.g. that $\{p_1, \ldots, p_N\}$ is a subset of the neurons with skip connections to the output layer and satisfy Assumption 3.1. In the following, we will show that there exists a weight configuration $U$ such that the submatrix $\Psi_{1:N,1:N}$ has full rank. Using then that the determinant is an analytic function together with Lemma A.1, we will conclude that the set of weight configurations $U$ such that $\Psi$ has *not* full rank has Lebesgue measure zero.

We remind that all the hidden units in the network are indexed from the first hidden layer till the higher layers as $d+1, \ldots, d+H$. For every hidden neuron $j \in [d+1, d+H]$, $u_j$ denotes the associated weight vector

$$u_j = [u_{k \to j}]_{k \in \text{in}(j)} \in \mathbb{R}^{|\text{in}(j)|}, \quad \text{where } \text{in}(j) = \text{the set of incoming units to unit } j.$$

Let $n_1$ be the number of units of the first hidden layer. For every neuron $j$ from the first hidden layer, let us define the pre-activation output $g_j$,

$$g_j(x_i) = \sum_{k \to j} (x_i)_k u_{k \to j}.$$

Due to Assumptions 3.1 (condition 3), we can always choose the weights $\{u_{d+1}, \ldots, u_{d+n_1}\}$ so that the output of every neuron in the first layer is distinct for different training samples, that is $g_j(x_i) \neq g_j(x_{i'})$ for every $j \in [d+1, d+n_1]$ and $i \neq i'$. For every neuron $j \in [d+n_1+1, d+H]$ in the higher layers we choose the weight vector $u_j$ such that it has exactly one 1 and 0 elsewhere. According to our definition of network in Section 2, the weight vectors of neurons of the same layer need not have the same dimension, but any subgroup of these neurons can still have shared weights as long as the dimensions among them agree. Thus the above choice of $u$ is always possible. In the following, let $c(j)$ denote the neuron below $j$ such that $u_{c(j) \to j} = 1$. This leads to

$$\sum_{k \to j} f_k(x) u_{k \to j} = f_{c(j)}(x).$$

Let $\alpha := (\alpha_{d+1}, \ldots, \alpha_{d+H})$ be a tuple of positive scalars. Let $\beta \in \mathbb{R}$ such that $\sigma_{p_j}(\beta) \neq 0$ for every $j \in [N]$. We consider a family of configurations of network parameters of the form $(\alpha_j u_j, b_j)_{j=d+1}^{d+H}$, where the biases are chosen as

$$b_{p_j} = \beta - \alpha_{p_j} g_{p_j}(x_j) \quad \forall j \in [N], p_j \in [d+1, d+n_1]$$
$$b_{p_j} = \beta - \alpha_{p_j} f_{c(p_j)}(x_j) \quad \forall j \in [N], p_j \notin [d+1, d+n_1]$$
$$b_j = 0 \quad \forall j \in \{d+1, \ldots, d+H\} \setminus \{p_1, \ldots, p_N\}$$

Note that the assignment of biases can be done via a forward pass through the network. By the above choice of biases and our definition of neurons in Section 2, we have

$$f_{p_j}(x_i) = \sigma_{p_j}\Big(\beta + \alpha_{p_j}\big(g_{p_j}(x_i) - g_{p_j}(x_j)\big)\Big), \quad \forall j \in [N], p_j \in [d+1, d+n_1],$$
$$f_{p_j}(x_i) = \sigma_{p_j}\Big(\beta + \alpha_{p_j}\big(f_{c(p_j)}(x_i) - f_{c(p_j)}(x_j)\big)\Big) \quad \forall j \in [N], p_j \notin [d+1, d+n_1],$$
$$f_j(x_i) = \sigma_j\Big(\alpha_j f_{c(j)}(x_i)\Big) \quad \forall j \in \{d+n_1+1, \ldots, d+H\} \setminus \{p_1, \ldots, p_N\},$$
$$f_j(x_i) = \sigma_j\Big(\alpha_j g_j(x_i)\Big) \quad \forall j \in \{d+1, \ldots, d+n_1\} \setminus \{p_1, \ldots, p_N\}. \tag{5}$$

One notes that the output of every skip-connection neuron $p_j$ is given by the first equation if $p_j$ lies on the first layer and by the second equation if $p_j$ lies on higher layers. In the following, to reduce notational complexity we make a convention that: $f_{c(p_j)} = g_{p_j}$ for every $p_j$ lies on the first layer. This allows us to use the second equation for every skip-connection neuron, that is,

$$f_{p_j}(x_i) = \sigma_{p_j}\Big(\beta + \alpha_{p_j}\big(f_{c(p_j)}(x_i) - f_{c(p_j)}(x_j)\big)\Big) \forall j \in [N]. \tag{6}$$

Now, since $\alpha > 0$ and all activation functions are strictly increasing by Assumption 3.1, one can easily show from the above recursive definitions that if $p_j$ is a skip-connection neuron which does

not lie on the first hidden layer then one has the relation: $f_{c(p_j)}(x_i) < f_{c(p_j)}(x_j)$ if and only if $g_{q_j}(x_i) < g_{q_j}(x_j)$, where $q_j$ is some neuron in the first hidden layer. This means if one sorts the elements of the set $\{f_{c(p_j)}(x_1), \ldots, f_{c(p_j)}(x_N)\}$ in increasing order then for every positive tuple $\alpha$, the order is fully determined by the corresponding order of $\{g_{q_j}(x_1), \ldots, g_{q_j}(x_N)\}$ for some neuron $q_j$ in the first layer. Note that this order can be different for different neurons $q_j$ in the first layer, and thus can be different for different skip-connection neurons $p_j$. Let $\pi$ be a permutation such that it holds for every $j = 1, 2, \ldots, N$ that

$$\pi(j) = \underset{i \in \{1, \ldots, N\} \setminus \{\pi(1), \ldots, \pi(j-1)\}}{\arg\max} f_{c(p_j)}(x_i) \tag{7}$$

It follows from above that $\pi$ is fully determined by the values of $g$ at the first layer. By definition one has $f_{c(p_j)}(x_{\pi_i}) < f_{c(p_j)}(x_{\pi_j})$ for every $i > j$. Since $\pi$ is independent of every positive tuple $\alpha$ and fully determined by the values of $g$, it can be fixed in the beginning. One can assume w.l.o.g. that $\pi$ is the identity permutation as otherwise one can reorder the training samples according to $\pi$ so that the rank of $\Psi$ does not change. Thus it holds for every $\alpha > 0$ that

$$\delta_{ij} := f_{c(p_j)}(x_i) - f_{c(p_j)}(x_j) < 0 \quad \forall i, j \in [N], i > j \tag{8}$$

Now, we are ready to show that there exists a positive tuple $\alpha$ for which $\Psi$ has full rank. We consider two cases of the activation functions of skip-connection neurons as stated in Assumption 3.1:

- In the first case, the activation functions $\sigma_{p_j} : \mathbb{R} \to \mathbb{R}$ for every $j \in [N]$ are strictly increasing, bounded and $\lim_{t \to -\infty} \sigma_{p_j}(t) = 0$. In the following, let $l(j)$ denote the layer index of the hidden unit $j$. For every hidden unit $j \in \{d+1, \ldots, d+H\}$ we set $\alpha_j$ to be the maximum of certain bounds (explained later in (10)) associated to all skip-connection neurons $p_k$ lying on the same layer, that is,

$$\alpha_j = \max \left\{ 1, \max_{k \in [N] | l(p_k) = l(j)} \max_{i > k} \frac{\sigma_{p_k}^{-1}(\epsilon) - \beta}{f_{c(p_k)}(x_i) - f_{c(p_k)}(x_k)} \right\} \tag{9}$$

  where $\epsilon > 0$ is an arbitrarily small constant which will be specified later. There are a few remarks we want to make for Eq. (9) before proceeding with our proof. First, the second term in (9) can be empty if there is no skip-connection unit $p_k$ which lies on the same layer as unit $j$, in which case $\alpha_j$ is simply set to 1. Second, $\alpha_j$'s are well-defined by constructing the values $f_{c(p_k)}(x_r)$, $r = 1, \ldots, N$ by a forward pass through the network (note that the network is a directed, acyclic graph; in particular, in the formula of $\alpha_j$, one has $l(c(p_k)) < l(p_k) = l(j)$ and thus the computation of $\alpha_j$ is feasible given the values of hidden units lying below the layer of unit $j$, namely $f_{c(p_k)}$). Third, if $j$ and $j'$ are two neurons from the same layer, *i.e.* $l(j) = l(j')$, then it follows from (9) that $\alpha_j = \alpha_{j'}$, meaning that their corresponding weight vectors are scaled by the same factor, thus any potential weight sharing conditions imposed on these neurons can still be satisfied.

  The main idea of choosing the above values of $\alpha$ is to obtain

$$\Psi_{ij} = f_{p_j}(x_i) \leq \epsilon \quad \forall i, j \in [N], i > j. \tag{10}$$

  To see this, one first observes that the inequality (8) holds for the constructed values of $\alpha$ since they are all positive. From (9) it holds for every skip-connection unit $p_j$ that

$$\alpha_{p_j} > \max_{i > j} \frac{\sigma_{p_j}^{-1}(\epsilon) - \beta}{f_{c(p_j)}(x_i) - f_{c(p_j)}(x_j)} \quad \forall j \in [N]$$

  which combined with (8) leads to

$$\alpha_{p_j}(f_{c(p_j)}(x_i) - f_{c(p_j)}(x_j)) \leq \sigma_{p_j}^{-1}(\epsilon) - \beta \quad \forall i, j \in [N], i > j.$$

  and thus using (6) we obtain (10).

  Coming back to the main proof of the lemma, since $\sigma_{p_j}(j \in [N])$ are bounded there exists a finite positive constant $C$ such that it holds that

$$|\Psi_{ij}| \leq C \quad \forall i, j \in [N] \tag{11}$$

By the Leibniz-formula one has

$$\det(\Psi_{1:N,1:N}) = \prod_{j=1}^{N} \sigma_{p_j}(\beta) + \sum_{\pi \in S_N \setminus \{\gamma\}} \text{sign}(\pi) \prod_{j=1}^{N} \Psi_{\pi(j)j} \qquad (12)$$

where $S_N$ is the set of all $N!$ permutations of the set $\{1, \dots, N\}$ and $\gamma$ is the identity permutation. Now, one observes that for every permutation $\pi \neq \gamma$, there always exists at least one component $j$ where $\pi(j) > j$ in which case it follows from (10) and (11) that

$$\Big| \sum_{\pi \in S_N \setminus \{\gamma\}} \text{sign}(\pi) \prod_{j=1}^{N} \Psi_{\pi(j)j} \Big| \leq N! \, C^{N-1} \epsilon$$

By choosing $\epsilon = \dfrac{\left| \prod_{j=1}^{N} \sigma_{p_j}(\beta) \right|}{2N! C^{N-1}}$, we get that

$$\det(\Psi_{1:N,1:N}) \geq \prod_{j=1}^{N} \sigma_{p_j}(\beta) - \frac{1}{2} \prod_{j=1}^{N} \sigma_{p_j}(\beta) = \frac{1}{2} \prod_{j=1}^{N} \sigma_{p_j}(\beta) \neq 0$$

and thus $\Psi$ has full rank.

- In the second case we consider the softplus activation function which satisfies our Assumption 3.1 that there exists a backward path from every skip-connection neuron $p_j$ to the first hidden layer s.t. on this path there is no neuron which has skip-connections to the output or shared weights with other skip-connection neurons.

  We choose all the weights and biases similarly to the first case. The only difference is that for every skip-connection neuron $p_j (1 \leq j \leq N)$, the position of 1 in its weight vector $u_{p_j}$ is chosen s.t. the value of neuron $p_j$ is determined by the first neuron on the corresponding backward path as stated in Assumption 3.1, that is,

$$\sum_{k \to p_j} f_k(x_i) u_{k \to p_j} = f_{c(p_j)}(x_i).$$

For skip-connection neurons we set all $\{\alpha_{p_1}, \dots, \alpha_{p_N}\}$ to some scalar variable $\alpha$, and for non-skip connection neurons $j$ we set $\alpha_j = 1$. From (6) and equations of (5) we have

$$f_{p_j}(x_i) = \sigma_{p_j}\Big(\beta + \alpha\big(f_{c(p_j)}(x_i) - f_{c(p_j)}(x_j)\big)\Big) \quad \forall j \in [N],$$
$$f_j(x_i) = \sigma_j\big(f_{c(j)}(x_i)\big) \quad \forall j \in \{d+1, \dots, d+H\} \setminus \{p_1, \dots, p_N\}. \qquad (13)$$

Note that with above construction of $u$ and $\alpha$, the only case where our weight sharing conditions can be potentially violated is between a skip-connection neuron ($\alpha_j = \alpha$) with a neuron on a backward path ($\alpha_j = 1$). However, this is not possible because our assumption in this case states that there is no weight sharing between a skip-connection neuron and a neuron on one of the backward paths.

Next, by our assumption the recursive backward path $c^{(k)}(p_j)$ does not contain any skip-connection unit and thus will eventually end up at some neuron $q_j \in [d+1, d+n_1]$ in the first hidden layer after some finite number of steps. Thus we can write for every $j \in [N]$

$$f_{p_j}(x_i) = \sigma_{p_j}\Big(\beta + \alpha\big(f_{c(p_j)}(x_i) - f_{c(p_j)}(x_j)\big)\Big),$$

where

$$f_{c(p_j)}(x_i) = \sigma_{c(p_j)}(\sigma_{c(c(p_j))}(\dots (g_{q_j}(x_i)) \dots)) \quad \forall i \in [N].$$

Moreover, we have from (8) that $f_{c(p_j)}(x_i) < f_{c(p_j)}(x_j)$ for every $i > j$. Note that softplus fulfills for $t < 0$, $\sigma_\gamma(t) \leq \frac{1}{\gamma} e^{\gamma t}$, whereas for $t > 0$ one has $\sigma_\gamma(t) \leq \frac{1}{\gamma} + t$. The latter property implies $\sigma^{(K)}(t) \leq \frac{K}{\gamma} + t$. Finally, this together implies that there exist positive constants $c_1, c_2, c_3, c_4$ such that it hods

$$|\prod_{j=1}^{N} \Psi_{\pi(j)j}| \leq c_1 e^{-\alpha c_2} (c_3 + \alpha)^{N-1}.$$

This can be made arbitrarily small by increasing $\alpha$. Thus we get

$$\lim_{\alpha \to \infty} \det(\Psi_{1:N,1:N}) = \prod_{j=1}^{N} \sigma_{p_j}(\beta) \neq 0$$

So far, we have shown that there always exist $U$ such that $\Psi$ has full rank. Since every activation function is real analytic by Assumption 3.1, every entry of $\Psi$ is also a real analytic function of the network parameters where $\Psi$ depends on. The set of low rank matrices $\Psi$ can be characterized by a system of equations such that all the $\binom{M}{N}$ determinants of all $N \times N$ sub-matrices of $\Psi$ are zero. As the determinant is a polynomial in the entries of the matrix and thus an analytic function of the entries and composition of analytic functions are again analytic, we conclude that each determinant is an analytic function of $U$. As shown above, there exists at least one $U$ such that one of these determinant functions is not identically zero and thus by Lemma A.1, the set of $U$ where this determinant is zero has measure zero. But as all submatrices need to have low rank in order that $\Psi$ has low rank, it follows that the set of $U$ where $\Psi$ has low rank has just measure zero. $\square$

## C    EXTENSION OF THEOREM 3.4 TO GENERAL CONVEX LOSSES

In this section, we consider a more general training objective, defined as

$$\Phi(U,V) = \varphi(G(U,V))$$

where $G(U,V) = \Psi(U)V \in \mathbb{R}^{N \times m}$ is the output of the network for all training samples at some given parameters $(U,V)$, and $\varphi : \mathbb{R}^{N \times m} \to \mathbb{R}$ the loss function applied on the network output.

**Assumption C.1** *The loss function $\varphi : \mathbb{R}^{N \times m} \to \mathbb{R}$ is convex and bounded from below.*

One can easily check that the following loss functions satisfy Assumption C.1 as they are all convex and bounded from below by zero:

1. The cross-entropy loss from Equation 2, in particular:

$$\varphi(G) = \frac{1}{N} \sum_{i=1}^{N} -\log\left(\frac{e^{G_{iy_i}}}{\sum_{k=1}^{m} e^{G_{ik}}}\right),$$

where $(x_i, y_i)_{i=1}^{N}$ is the training data with $y_i$ being the ground-truth class of $x_i$.

2. The standard square loss (for classification/regression tasks)

$$\varphi(G) = \frac{1}{2} \|G - Y\|_F^2, \tag{14}$$

where $Y \in \mathbb{R}^{N \times m}$ is the ground-truth matrix.

3. The multi-class Hinge-loss (for classification tasks)

$$\varphi(G) = \frac{1}{N} \sum_{i=1}^{N} \max_{j \neq y_i} \max(0, 1 - (G_{iy_i} - G_{ij})),$$

where $(x_i, y_i)_{i=1}^{N}$ is the training data with $y_i$ being the ground-truth class of $x_i$.

By Assumption C.1, $\varphi$ is bounded from below, thus it attains a finite infimum:

$$p^* := \inf_{G \in \mathbb{R}^{N \times m}} \varphi(G) < \infty.$$

Basically, $p^*$ serves as a lower bound on our training objective $\Phi$. For the above examples, it holds $p^* = 0$. Next, we adapt the definition of "bad local valleys" from Definition 3.3 to the current setting.

**Definition C.2** *The $\alpha$-sublevel set of $\Phi$ is defined as $L_\alpha = \{(U, V) \mid \Phi(U, V) < \alpha\}$. A local valley is defined as a connected component of some sublevel set $L_\alpha$. A bad local valley is a local valley on which the training objective $\Phi$ cannot be made arbitrarily close to $p^*$.*

The following result extends Theorem 3.4 to general convex losses. The proofs are mostly similar as before, but we present them below for the completeness and convenience of the reader.

**Theorem C.3** *The following holds under Assumption 3.1 and Assumption C.1:*

1. *There exist uncountably many solutions with zero training error.*

2. *The loss landscape of $\Phi$ does not have any bad local valley.*

3. *There exists no suboptimal strict local minimum.*

4. *For cross-entropy loss* (2) *and square loss* (14) *there exists no local maximum.*

**Proof:**

1. By Lemma 3.2 the set of $U$ such that $\Psi(U)$ has not full rank $N$ has Lebesgue measure zero. Given $U$ such that $\Psi$ has full rank, the linear system $\Psi(U)V = Y$ has for every possible target output matrix $Y \in \mathbb{R}^{N \times m}$ at least one solution $V$. As this is possible for almost all $U$, there exist uncountably many solutions achieving zero training error.

2. Let $C$ be a non-empty, connected component of some sub-level set $L_\alpha$ where $\alpha > p^*$. Note that $L_\alpha = \varnothing$ if $\alpha \leq p^*$ by Definition C.2. Given any $\epsilon \in (p^*, \alpha)$, we will show that $C$ always contains a point $(U, V)$ s.t. $\Phi(U, V) \leq \epsilon$ as this would imply that the loss $\Phi$ restricted to $C$ can always attain arbitrarily small value close to $p^*$.

   We note that $L_\alpha = \Phi^{-1}((-\infty, \alpha))$ is an open set according to Proposition A.2. Since $C$ is a non-empty connected component of $L_\alpha$, $C$ must also be an open set with non-zero Lebesgue measure. By Lemma 3.2 the set of $U$ where $\Psi(U)$ has not full rank has measure zero and thus $C$ must contain a point $(U, V)$ such that $\Psi(U)$ has full rank. By Assumption C.1, $\varphi$ attains its infimum at $p^* < \epsilon$, and thus by continuity of $\varphi$, there exists $G^* \in \mathbb{R}^{N \times m}$ such that $p^* \leq \varphi(G^*) \leq \epsilon$. As $\Psi(U)$ has full rank, there always exist $V^*$ such that $\Psi(U)V^* = G^*$. Now, one notes that the loss $\Phi(U, V) = \varphi(\Psi(U)V)$ is convex in $V$, and that $\Phi(U, V) < \alpha$, thus we have for the line segment $V(\lambda) = \lambda V + (1 - \lambda)V^*$ for $\lambda \in [0, 1]$,

$$\Phi(U, V(\lambda)) \leq \lambda \Phi(U, V) + (1 - \lambda)\Phi(U, V^*) < \lambda \alpha + (1 - \lambda)\epsilon < \alpha.$$

   Thus the whole line segment from $(U, V)$ to $(U, V^*)$ is contained in $L_\alpha$. Since $C$ is a connected component of $L_\alpha$ which contains $(U, V)$, it follows that $(U, V^*) \in C$. Moreover, one has $\Phi(U, V^*) = \varphi(G^*) \leq \epsilon$, which thus implies that the loss can always be made $\epsilon$-small inside the set $C$ for every $\epsilon \in (p^*, \alpha)$.

3. Let $(U_0, V_0)$ be a strict suboptimal local minimum, then there exists $r > 0$ such that $\Phi(U, V) > \Phi(U_0, V_0) > p^*$ for all $(U, V) \in B((U_0, V_0), r) \setminus \{(U_0, V_0)\}$ where $B(\cdot, r)$ denotes a closed ball of radius $r$. Let $\alpha = \min_{(U,V) \in \partial B((U_0, V_0), r)} \Phi(U, V)$ which exists as $\Phi$ is continuous and the boundary $\partial B((U_0, V_0), r)$ of $B((U_0, V_0), r)$ is compact. Note that $\Phi(U_0, V_0) < \alpha$ as $(U_0, V_0)$ is a strict local minimum, and thus $(U_0, V_0) \in L_\alpha$. Let $E$ be the connected component of $L_\alpha$ which contains $(U_0, V_0)$, that is, $(U_0, V_0) \in E \subseteq L_\alpha$. Since the loss of every point inside $E$ is strictly smaller than $\alpha$, whereas the loss of every point on the boundary $\partial B((U_0, V_0), r)$ is greater than or equal to $\alpha$, $E$ must be contained in the interior of the ball, that is $E \subset B((U_0, V_0), r)$. Moreover, $\Phi(U, V) \geq \Phi(U_0, V_0) > p^*$ for every $(U, V) \in E$ and thus the values of $\Phi$ restricted to $E$ can not be arbitrarily close to $p^*$, which means that $E$ is a bad local valley, which contradicts G.3.2.

4. The proof for cross-entropy loss is similar to Theorem 3.4. For square loss, one has

$$\Phi(U, V) = \frac{1}{2} \|\Psi(U)V - Y\|_F^2 = \frac{1}{2} \|(\mathbb{I}_m \otimes \Psi(U)) \, vec(V) - vec(Y)\|_2^2$$

   where $\otimes$ denotes Kronecker product, and $\mathbb{I}_m$ an $m \times m$ identity matrix. The hessian of $\Phi$ w.r.t. $V$ is $\nabla_{vec(V)}^2 \Phi = (\mathbb{I}_m \otimes \Psi(U))^T (\mathbb{I}_m \otimes \Psi(U))$.

Suppose by contradiction that $(U, V)$ is a local maximum. Then the Hessian of $\Phi$ is negative semi-definite. As principal submatrices of negative semi-definite matrices are again negative semi-definite, the Hessian of $\Phi$ w.r.t $V$ must be also negative semi-definite. However, $\Phi$ is convex in $V$ thus its Hessian restricted to $V$ must be positive semi-definite. The only matrix which is both p.s.d. and n.s.d. is the zero matrix. It follows that $\nabla^2_{vec(V)}\Phi(U, V) = 0$ and thus $\Psi(U) = 0$. By Assumption 3.1, there exists $j \in [M]$ s.t. $\sigma_{p_j}$ is strictly positive, thus some entries of $\Psi(U)$ must be strictly positive, and so $\Psi(U)$ cannot be identically zero, leading to a contradiction. Therefore $\Phi$ has no local maximum.

$\square$

## D   THE ARCHITECTURE OF CNN13 FROM TABLE 1: SEE TABLE 3

| Layer | Output size | #neurons |
|---|---|---|
| Input: $28 \times 28$ | $28 \times 28 \times 1$ | |
| $3 \times 3\,\mathrm{conv} - 64,\ \ \mathrm{stride}\,1$ | $28 \times 28 \times 64$ | 50176 |
| $3 \times 3\,\mathrm{conv} - 64,\ \ \mathrm{stride}\,1$ | $28 \times 28 \times 64$ | 50176 |
| $3 \times 3\,\mathrm{conv} - 64,\ \ \mathrm{stride}\,2$ | $14 \times 14 \times 64$ | 12544 |
| $3 \times 3\,\mathrm{conv} - 128,\ \mathrm{stride}\,1$ | $14 \times 14 \times 128$ | 25088 |
| $3 \times 3\,\mathrm{conv} - 128,\ \mathrm{stride}\,1$ | $14 \times 14 \times 128$ | 25088 |
| $3 \times 3\,\mathrm{conv} - 128,\ \mathrm{stride}\,2$ | $7 \times 7 \times 128$ | 6272 |
| $3 \times 3\,\mathrm{conv} - 256,\ \mathrm{stride}\,1$ | $7 \times 7 \times 256$ | 12544 |
| $1 \times 1\,\mathrm{conv} - 256,\ \mathrm{stride}\,1$ | $7 \times 7 \times 256$ | 12544 |
| $3 \times 3\,\mathrm{conv} - 256,\ \mathrm{stride}\,2$ | $4 \times 4 \times 256$ | 4096 |
| $3 \times 3\,\mathrm{conv} - 256,\ \mathrm{stride}\,1$ | $4 \times 4 \times 256$ | 4096 |
| $3 \times 3\,\mathrm{conv} - 256,\ \mathrm{stride}\,2$ | $2 \times 2 \times 256$ | 1024 |
| $3 \times 3\,\mathrm{conv} - 256,\ \mathrm{stride}\,1$ | $2 \times 2 \times 256$ | 1024 |
| $3 \times 3\,\mathrm{conv} - 256,\ \mathrm{stride}\,2$ | $1 \times 1 \times 256$ | 256 |
| Fully connected, 10 output units | | |

Table 3: The architecture of CNN13 for MNIST dataset. There are in total $179,840$ hidden neurons.

## E   VISUALIZATION OF THE LOSS LANDSCAPE BEFORE AND AFTER ADDING SKIP-CONNECTIONS TO THE OUTPUT LAYER

Similar to Li et al. (2018); Goodfellow et al. (2015), we visualize the loss surface restricted to a two dimensional subspace of the parameter space. The subspace is chosen to go through some point $(U_0, V_0)$ learned by SGD and spanned by two random directions $(U_1, V_1)$ and $(U_2, V_2)$.

For the purpose of illustration, we train with SGD a two-hidden-layer fully connected network with 784 and 300 hidden units respectively, followed by a 10-way softmax. The training set consists of 1024 images, which are randomly selected from MNIST dataset. After adding skip-connections to the output, the network fulfills $M = N = 1024$. Figure 3 shows the heat map of the loss surface before and after adding skip-connections. One can see a visible effect that skip-connections have helped to smooth the loss landscape near a small sub-optimal region and allows gradient descent to flow directly from there to the bottom of the landscape with smaller objective value.

## F   DISCUSSION OF TRAINING ERROR IN TABLE 2

**Training error for the experiment in Table 2.**   As shown in Table 2, the training error is zero in all cases, except when the original VGG models are used with sigmoid activation function. The reason, as noticed in our experiments, is because the learning of these sigmoidal networks converges quickly to a constant zero classifier (i.e. the output of the last hidden layer converges to zero), which makes both training and test accuracy converge to $10\%$ and the loss in Equation (2) converges to

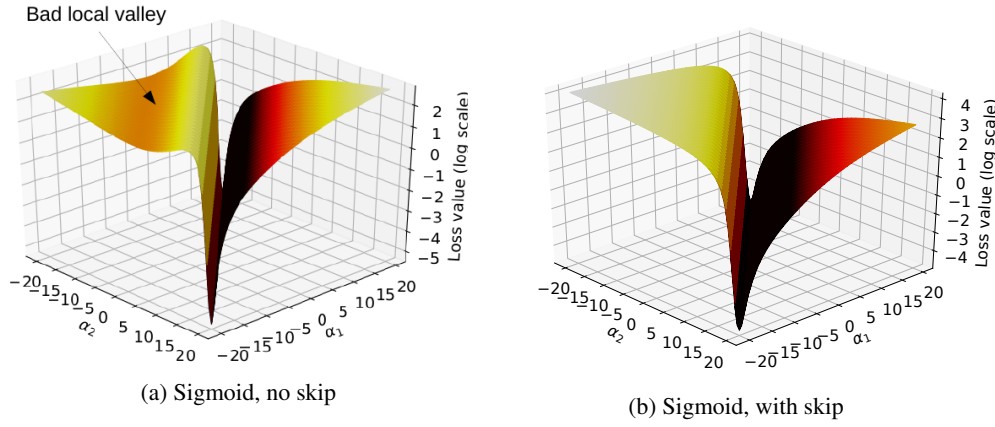

(a) Sigmoid, no skip    (b) Sigmoid, with skip

Figure 3: Loss surface of a two-hidden-layer network on a small MNIST dataset.

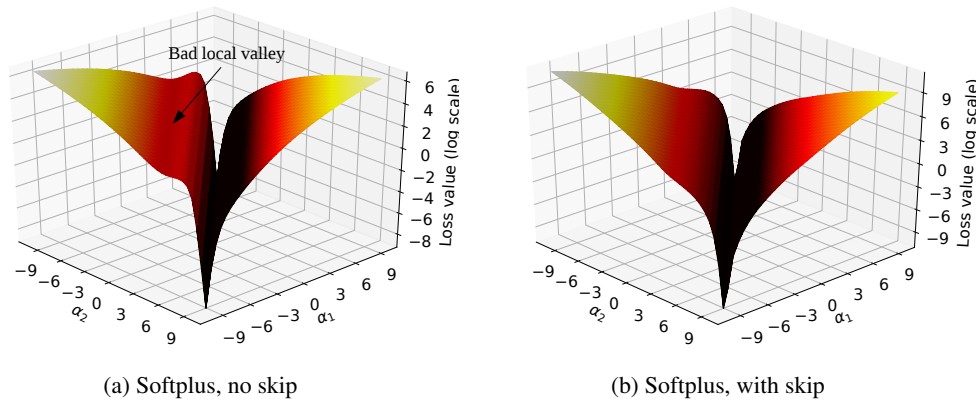

(a) Softplus, no skip    (b) Softplus, with skip

$-\log(1/10)$. While we are not aware of a theoretical explanation for this behavior, it is not restricted to the specific architecture of VGGs but hold in general for plain sigmoidal networks with depth>5 as pointed out earlier by Glorot & Bengio (2010). As shown in Table 2, Densenets however do not suffer from this phenomenon, probably because they already have skip-connections between all the hidden layers of a dense block, thus gradients can easily flow from the output to every layer of a dense block, which makes the training of this network with sigmoid activation function become feasible.

**Discussion of convergence speed.** For sigmoid activation, we noticed that skip-models when trained with the random sampling approach (skip-rand) often converge much slower than when trained with full SGD (skip-SGD). In our experiments, to be sure that one gets absolute zero training error, we set the number of training epochs to 5000 for the former case and 1000 for the later. Perhaps a better learning rate schedule might help to reduce this number, or maybe not, but this is beyond the scope of this paper. For softplus activation, we noticed a much faster convergence – all models often converge within 300 epochs to absolute zero training error.

**Skip-connections are also helpful for training very deep networks with softplus activation.** Previously we have shown that skip-connections are helpful for training deep sigmoidal networks. In this part, we show a similar result for softplus activation function. For the purpose of illustration, we create a small dataset with $N = 1000$ training images randomly chosen from CIFAR10 dataset. We use a very deep network with 150 fully connected layers, each of width 10, and softplus activation. A skip-model is created by adding skip-connections from $N$ randomly chosen neurons to the output units. We train both networks with SGD. The best learning rate for each model is empirically chosen from $\left\{10^{-2}, 10^{-3}, 10^{-4}, 10^{-5}\right\}$. We report the training loss and training error of both models in Figure 5. One can see that the skip-network easily converge to zero training error within 200 epochs, whereas the original network has stronger fluctuations and fails to converge after 1000 epochs. This

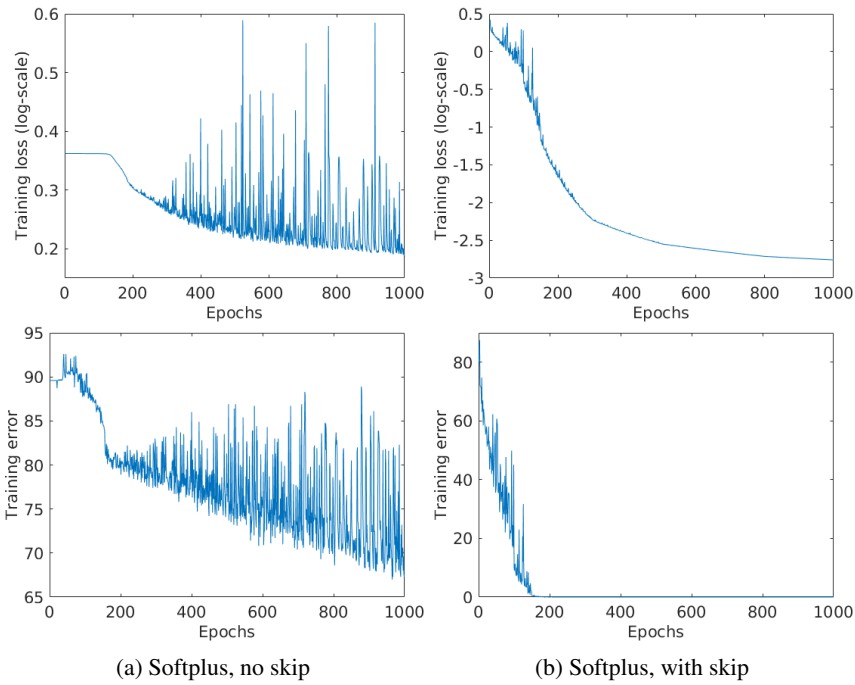

(a) Softplus, no skip        (b) Softplus, with skip

Figure 5: Training progress of a 150-layer neural network with and without skip-connections.

is directly related to our result of Theorem 3.4 in the sense that skip-connections can help to smooth the loss landscape and enable effective training of very deep networks.

## G  ADDITIONAL EXPERIMENTS: MAX-POOLING OF VGGS ARE REPLACED BY 2X2 CONVOLUTIONAL LAYERS OF STRIDE 2

The original VGG Simonyan & Zisserman (2015) and Densenet Huang et al. (2017) contain pooling layers in their architecture. In particular, original VGGs have max-pooling layers, and original Densenets have averaging pooling layers. In the following, we will clarify how/if these pooling layers have been used in our experiments in Table 2, and whether and how our theretical results are appicable to this case, as well as presenting additional experimental results in this regard.

First of all, we note that Densenets Huang et al. (2017) contain pooling layers only after the first dense block. Meanwhile, as noted in Table 2, our experiments with Densenets only use skip-connections from hidden units of the first dense block, and thus Lemma 3.2 and Theorem 3.4 are applicable. The reason is that one can restrict the full-rank analysis of matrix $\Psi$ in Lemma 3.2 to the hidden units of the first dense block, so that it follows that the set of parameters of the first dense block where $\Psi$ has not full rank has Lebesgue measure zero, from which our results of Theorem 3.4 follow immediately.

However for VGGs in Table 2, we kept their max-pooling layers similar to the original architecture as we wanted to have a fair comparison between our skip-models and the original models. In this setting, our results are not directly applicable because we lose the analytic property of the entries of $\Psi$ w.r.t. its dependent parameters, which is crucial to prove Lemma 3.2. Therefore in this section, we would like to present additional results to Table 2 in which we replace all max-pooling layers of all VGG models from Table 2 with 2x2 convolutional layers of stride 2. In this case, the whole network consists of only convolutional and fully connected layers, hence our theoretical results are applicable.

The experimental results are presented in Table 4. Overall, our main observations are similar as before. The performance gap between original models and their corresponding skip-variants are approximately the same as in Table 2 or slightly more pronounced in some cases. A one-to-one comparison with Table 2 also shows that the performance of skip-models themselves have decreased by $4-7\%$ after the replacement of max-pooling layers with 2x2 convolutional layers. This is perhaps

| | Sigmoid activation function | | Softplus activation function | |
|---|---|---|---|---|
| **Model** | C-10 | C-10$^+$ | C-10 | C-10$^+$ |
| VGG11-mp2conv | 10 | 10 | 74.53 | 88.80 |
| VGG11-mp2conv-skip (rand) | $53.65 \pm 0.66$ | - | $55.51 \pm 0.43$ | - |
| VGG11-mp2conv-skip (SGD) | $\mathbf{64.45} \pm 0.31$ | $\mathbf{80.15} \pm 0.59$ | $\mathbf{76.18} \pm 0.58$ | $\mathbf{89.93} \pm 0.19$ |
| VGG13-mp2conv | 10 | 10 | 74.04 | 90.37 |
| VGG13-mp2conv-skip (rand) | $53.45 \pm 0.23$ | - | $53.33 \pm 0.67$ | - |
| VGG13-mp2conv-skip (SGD) | $\mathbf{63.53} \pm 0.37$ | $\mathbf{82.40} \pm 0.23$ | $\mathbf{75.58} \pm 0.77$ | $\mathbf{91.04} \pm 0.20$ |
| VGG16-mp2conv | 10 | 10 | 74.00 | 90.38 |
| VGG16-mp2conv-skip (rand) | $53.83 \pm 0.30$ | - | $55.34 \pm 0.64$ | - |
| VGG16-mp2conv-skip (SGD) | $\mathbf{65.77} \pm 0.67$ | $\mathbf{83.06} \pm 0.34$ | $\mathbf{76.52} \pm 0.78$ | $\mathbf{91.00} \pm 0.23$ |

Table 4: Test accuracy (%) of VGG networks from Table 2 where max-pooling layers are replaced by 2x2 convolutional layers of stride 2 (denoted as mp2conv). Other notations are similar to Table 2.

not so surprising because the problem gets potentially harder when the network has more layers to be learned, especially in case of sigmoid activation where the decrease is sharper. Similar to Table 2, adding skip-connections to the output units still prove to be very helpful – it improves the result for softplus while making the training of deep networks with sigmoid activation become possible at all. Finally, the training of full network with SGD still yields significantly better solutions in terms of generalization error than the random feature approach. This confirms once again the implicit bias of SGD towards high quality solutions among infinitely many solutions with zero training error.

## H    DATA-AUGMENTATION RESULTS FOR TABLE 2

The following Table 5 shows additional results to Table 2 where data-augmentation is used now. For data-augmentation, we follow the procedure as described in (Zagoruyko & Komodakis, 2016) by considering random crops of size $32 \times 32$ after $4$ pixel padding on each side of the training images and random horizontal flips with probability $0.5$. For the convenience of the reader, we also repeat the results of Table 2 in the new table 5.

| | Sigmoid activation function | | Softplus activation function | |
|---|---|---|---|---|
| **Model** | C-10 | C-10$^+$ | C-10 | C-10$^+$ |
| VGG11 | 10 | 10 | 78.92 | 88.62 |
| VGG11-skip (rand) | $62.81 \pm 0.39$ | - | $64.49 \pm 0.38$ | - |
| VGG11-skip (SGD) | $\mathbf{72.51} \pm 0.35$ | $\mathbf{85.55} \pm 0.09$ | $\mathbf{80.57} \pm 0.40$ | $\mathbf{89.32} \pm 0.16$ |
| VGG13 | 10 | 10 | 80.84 | 90.58 |
| VGG13-skip (rand) | $61.50 \pm 0.34$ | - | $61.42 \pm 0.40$ | - |
| VGG13-skip (SGD) | $\mathbf{70.24} \pm 0.39$ | $\mathbf{86.48} \pm 0.32$ | $\mathbf{81.94} \pm 0.40$ | $\mathbf{91.06} \pm 0.12$ |
| VGG16 | 10 | 10 | 81.33 | 90.68 |
| VGG16-skip (rand) | $61.57 \pm 0.41$ | - | $61.46 \pm 0.34$ | - |
| VGG16-skip (SGD) | $\mathbf{70.61} \pm 0.36$ | $\mathbf{86.42} \pm 0.31$ | $\mathbf{81.91} \pm 0.24$ | $\mathbf{91.00} \pm 0.22$ |
| Densenet121 | **86.41** | **90.93** | **89.31** | **94.20** |
| Densenet121-skip (rand) | $52.07 \pm 0.48$ | - | $55.39 \pm 0.48$ | - |
| Densenet121-skip (SGD) | $81.47 \pm 1.03$ | $90.32 \pm 0.50$ | $86.76 \pm 0.49$ | $93.23 \pm 0.42$ |

Table 5: Test accuracy (%) of several CNN architectures with/without skip-connections on CIFAR10 ($^+$ denotes data augmentation). For each model A, A-skip denotes the corresponding skip-model in which a subset of $N$ hidden neurons "randomly selected" from the hidden layers are connected to the output units. For Densenet121, these neurons are randomly chosen from the first dense block. The names in open brackets (rand/SGD) specify how the networks are trained: **rand** ($U$ is randomized and fixed while $V$ is learned with SGD), **SGD** (both $U$ and $V$ are optimized with SGD).

