# OpenReview forum: "On the loss landscape of a class of deep neural networks with no bad local valleys"
_ICLR.cc/2019/Conference_

### Official Review · AnonReviewer1 · 2018-11-02
**Interesting experimental results, but less significant theoretical contribution**

**Rating:** 6
**Confidence:** 5

**Review:**

This paper presents a class of neural networks that does not have bad local valleys. The “no bad local valleys” implies that for any point on the loss surface there exists a continuous path starting from it, on which the loss doesn’t increase and gets arbitrarily smaller and close to zero. The key idea is to add direct skip connections from hidden nodes (from any hidden layer) to the output.

The good property of loss surface for networks with skip connections is impressive and the authors present interesting experimental results pointing out that
* adding skip connections doesn’t harm the generalization.
* adding skip connections sometimes enables training for networks with sigmoid activation functions, while the networks without skip connections fail to achieve reasonable performance.
* comparison of the generalization performance for the random sampling algorithm vs SGD and its connection to implicit bias is interesting.

However, from a theoretical point of view, I would say the contribution of this work doesn’t seem to be very significant, for the following reasons:
* In the first place, figuring out “why existing models work” would be more meaningful than suggesting a new architecture which is on par with existing ones, unless one can show a significant performance improvement over the other ones.
* The proof of the main theorem (Thm 3.3) is not very interesting, nor develops novel proof techniques. It heavily relies on Lemma 3.2, which I think is the main technical contribution of this paper. Apart from its technicality in the proof, the statement of Lemma 3.2 is just as expected and gives me little surprise, because having more than N hidden nodes connected directly to the output looks morally “equivalent” to having a layer as wide as N, and it is known that in such settings (e.g. Nguyen & Hein 17’) it is easy to attain global minima.
* I also think that having more than N skip connections can be problematic if N is very large, for example N>10^6. Then the network requires at least 1M nodes to fall in this class of networks without bad local valleys. If it is possible to remove this N-hidden-node requirement, it will be much more impressive.

Below, I’ll list specific comments/questions about the paper.
* Assumption 3.1.2 doesn’t make sense. Assumption 3.1.2 says “there exists N neurons satisfying…” and then the first bullet point says “for all j = 1, …, M”. Also, the statement “one of the following conditions” is unclear. Does it mean that we must have either “N satisfying the first bullet” or “N satisfying the second bullet”, or does it mean we can have N/2 satisfying the first and N/2 satisfying the second?
* The paper does not describe where the assumptions are used. They are never used in the proof of Theorem 3.3, are they? I believe that they are used in the proof of Lemma 3.2 in the appendix, but if you can sketch/mention how the assumptions come into play in the proofs, that will be more helpful in understanding the meaning of the assumptions.
* Are there any specific reasons for considering cross-entropy loss only? Lemma 3.2 looks general, so this result seems to be applicable to other losses. I wonder if there is any difficulty with different losses.
* Are hidden nodes with skip connections connected to ALL m output nodes or just some of the output nodes? I think it’s implicitly assumed in the proof that they are connected to all output nodes, but in this case Figure 2 is a bit misleading because there are hidden nodes with skip connections to only one of the output nodes.
* For the experiments, how did you deal with pooling layers in the VGG and DenseNet architectures? Does max-pooling satisfy the assumptions? Or the experimental setting doesn’t necessarily satisfy the assumptions?
* Can you show the “improvement” of loss surface by adding skip connections? Maybe coming up with a toy dataset and network WITH bad local valleys will be sufficient, because after adding N skip connections the network will be free of bad local valleys.

Minor points
* In the Assumption 3.1.3, the $N$ in $r \neq s \in N$ means $[N]$?
* In the introduction, there is a sentence “potentially has many local minima, even for simple models like deep linear networks (Kawaguchi, 2016),” which is not true. Deep linear networks have only global minima and saddle points, even for general differentiable convex losses (Laurent & von Brecht 18’ and Yun et al. 18’).
* Assumption 3.1.3 looked a bit confusing to me at first glance. You might want to add some clarification such as “for example, in the fully connected network case, this means that all data points are distinct.”

---

> ### Author Response · Authors · 2018-11-17
> **Response to AnonReviewer1. Part 1**
>
> Thank you very much for the detailed feedbacks. Below are answers to your comments/questions in the order that they appear.
>
> * "In the first place, figuring out “why existing models work” would be more meaningful than suggesting a new architecture which is on par with existing ones, unless one can show a significant performance improvement over the other ones."
>
> We absolutely agree that understanding why existing models work is what one desires to achieve in the end. But to reach that point, one has to start somewhere, and make progress continually. This is the reason for the existence of a bunch of recent work on this topic:
>
> A. Choromanska, M. Hena, M. Mathieu, G. B. Arous, and Y. LeCun. The loss surfaces of multilayer networks. 2015.
> I. Safran and O. Shamir. On the quality of the initial basin in overspecified networks. 2016.
> B. D. Haeffele and R. Vidal. Global optimality in neural network training. 2017.
> H. Lu, K. Kawaguchi. Depth creates no bad local minima. 2017.
> M. Hardt and T. Ma. Identity matters in deep learning. 2017.
> C. Yun, S. Sra, and A. Jadbabaie. Global optimality conditions for deep neural networks. 2017.
> D. Soudry and E. Hoffer.  Exponentially vanishing sub-optimal local minima in multilayer neural networks. 2017.
> M. Nouiehed and M. Razaviyayn. Learning Deep Models: Critical Points and Local Openness. 2018.
> T. Laurent and J. H. von Brecht. The Multilinear Structure of ReLU Networks. 2018.
> S. Liang, R. Sun, J. D. Lee, and R. Srikant. Adding one neuron can eliminate all bad local minima. 2018.
>
> At the moment, we are not aware of any previous work which can prove directly strong theoretical results on the loss landscape of "existing models" which actually work in practice. Moreover in this paper, we show that the presented class of networks enjoy both strong theoretical properties and good empirical performance. We do not make great claim about the result, but we believe that this is a significant contribution to the literature, especially w.r.t. the recent great effort of the community in trying to make progress on theoretical understanding of deep learning models.
>
> * "The proof of the main theorem (Thm 3.3) is not very interesting, nor develops novel proof techniques. It heavily relies on Lemma 3.2, which I think is the main technical contribution of this paper. Apart from its technicality in the proof, the statement of Lemma 3.2 is just as expected and gives me little surprise, because having more than N hidden nodes connected directly to the output looks morally “equivalent” to having a layer as wide as N, and it is known that in such settings (e.g. Nguyen & Hein 17’) it is easy to attain global minima."
>
> The proof of our main result is simple and elegant, as also noted by AnonReviewer2. Simple proofs are often generalizable better to complex models. Thus we think that it is actually an advantage of this work.
> Can the reviewer elaborate on why the statement of Lemma 3.2 is just as expected? Given that said, does the reviewer have in mind an easier proof for this lemma? - which we would be very happy to know We would like to note that the class of networks analyzed in this Lemma is quite general and hence the mathematical proof is non-trivial.  We agree that one can view the N skip-connections as an implicit wide layer, but this is just an intuition and very weak argument to conclude that the statements are just as expected. There are things that might look "intuitive" and "as expected" but it's completely wrong, for instance, a deep linear network with N skip-connections to the output does not satisfy our conditions and results if the training data has very low rank.
>
> * "I also think that having more than N skip connections can be problematic if N is very large, for example N>10^6. Then the network requires at least 1M nodes to fall in this class of networks without bad local valleys. If it is possible to remove this N-hidden-node requirement, it will be much more impressive."
>
> We agree that the current condition on the number of skip-connections is quite strong. But on the other hand, it's not necessarily too restrictive at the level as mentioned by the reviewer. We would like to refer to Table 1 in [1] for some information on the number of neurons of the first layer of several existing networks. For instance, the first hidden layer of original VGG-Nets has already more than 3M nodes, and so if one sum up this number for all the hidden layers the total will be much than that. Moreover, in the literature it is common to find theoretical work which requires extremely larger number of neurons than the number of training samples, see e.g. https://openreview.net/forum?id=S1eK3i09YQ which requires N^6 neurons for gradient descent to find a zero training error solution for one hidden layer networks. Nevertheless, we agree with the reviewer that it would be interesting to relax this condition in future work.
>
> [1] Nguyen & Hein. Optimization landscape and expressivity of deep cnns. 2017.

---

> > ### Comment · AnonReviewer1 · 2018-11-19
> > **Reply to the response**
> >
> > First of all, I would like to appreciate the authors for their extensive efforts in revising and improving the paper.
> >
> > I think most of my concerns were more or less addressed, except for the “assumptions” and “proof technique” parts.
> >
> > First of all, I still believe it is weird that the assumptions are never used *explicitly* anywhere in the main text. The paper makes some assumptions and never uses them directly in the main text. I would suggest the authors to at least add a “proof sketch” paragraph below Lemma 3.2, and briefly outline the proof while mentioning how the assumptions come into play.
> >
> > As for the proof technique part, by “as expected” I meant I would have been more surprised if the set of U with rank-deficient \Psi(U) had measure greater than zero. This was because in general, rank deficient matrices lie in a set of measure zero, and I’ve seen many results such as “if a hidden layer is wider than N and activation functions have good properties, then some matrix has full rank almost everywhere.”
> >
> > Unfortunately, however, I can hardly agree that the proof is “elegant” at the moment, especially for Lemma 3.2. There are many steps that makes the proof unnecessarily longer. For example, the very first equation in step 1 is not necessary; you can just start with eq (4). Similarly, I believe that steps 2-5 can be made much more concise. In defining eq (9), why don’t you just start by “for all nodes j in layer l, define all \alpha_j to be:”? I also don’t understand a few lines above eq (10). Given that the network is not fully connected but a DAG, how can you guarantee that u_j and u_{j’} are of the same size and make them identical? For the softplus case, the choice of \beta is missing. Without this, how can you make sure that some of the data points fall into the negative side of softplus?
> >
> > I agree that there are some interesting techniques used in constructing the parameter U. However, the main theoretical contribution (proof of Lemma 3.2) is hidden in the appendix, which many readers will end up skipping. My current score is based on the main text, and at least in my opinion, the main text itself doesn’t reveal anything particularly interesting.

---

> > > ### Author Response · Authors · 2018-11-20
> > > **Thank you for your feedback**
> > >
> > > Thank you for the quick response.
> > >
> > > "First of all, I still believe it is weird that the assumptions are never used *explicitly* anywhere in the main text. The paper makes some assumptions and never uses them directly in the main text. I would suggest the authors to at least add a “proof sketch” paragraph below Lemma 3.2, and briefly outline the proof while mentioning how the assumptions come into play."
> > >
> > > We agree. We have added a proof sketch for Lemma 3.2 and briefly discussed how the assumptions are used now.
> > >
> > > "As for the proof technique part, by “as expected” I meant I would have been more surprised if the set of U with rank-deficient \Psi(U) had measure greater than zero. This was because in general, rank deficient matrices lie in a set of measure zero, and I’ve seen many results such as “if a hidden layer is wider than N and activation functions have good properties, then some matrix has full rank almost everywhere.”"
> > >
> > > Sure. But the problem becomes highly non-trivial when the matrix has very special and sophisticated structure, such as the one analyzed in this paper. Despite of all intuitions, it's still a mathematical problem that needs to be rigorously proved.
> > >
> > > "Unfortunately, however, I can hardly agree that the proof is “elegant” at the moment, especially for Lemma 3.2. There are many steps that makes the proof unnecessarily longer. For example, the very first equation in step 1 is not necessary; you can just start with eq (4). Similarly, I believe that steps 2-5 can be made much more concise. In defining eq (9), why don’t you just start by “for all nodes j in layer l, define all \alpha_j to be:”? I also don’t understand a few lines above eq (10). Given that the network is not fully connected but a DAG, how can you guarantee that u_j and u_{j’} are of the same size and make them identical? For the softplus case, the choice of \beta is missing. Without this, how can you make sure that some of the data points fall into the negative side of softplus?"
> > >
> > > Following reviewer's suggestion, we have revised/shortened the proof of Lemma 3.2. Please check our revision. Regarding u_j and u_{j'}, we already added further explanation in the proof. Basically they need not have the same size because according to our network description in Section 2, only those neurons with the same number of incoming units can have shared weights. For instance, it's fine to have on the same layer two neurons with weights (1,0,0) and other two neurons (0,0,1,0). The bias for softplus is mentioned now. The \beta variable is defined in the beginning, so basically we use the same value of \beta as in the first case.

---

> > > > ### Comment · AnonReviewer1 · 2018-11-27
> > > > **Thanks for your efforts**
> > > >
> > > > I appreciate the authors for their efforts in revising the paper. Many of my concerns are addressed throughout the revision/feedback process, and I think the paper is now in a better shape.
> > > >
> > > > I'll edit the rating accordingly.

---

> > > > > ### Author Response · Authors · 2018-11-27
> > > > > **Thank you**
> > > > >
> > > > > Thank you very much for your positive feedback and all the helpful comments so far.

---

> ### Author Response · Authors · 2018-11-17
> **Response to AnonReviewer1. Part 2**
>
> Answers to specific comments/questions:
> * "Assumption 3.1.2 doesn’t make sense. Assumption 3.1.2 says “there exists N neurons satisfying…” and then the first bullet point says “for all j = 1, …, M”. Also, the statement “one of the following conditions” is unclear. Does it mean that we must have either “N satisfying the first bullet” or “N satisfying the second bullet”, or does it mean we can have N/2 satisfying the first and N/2 satisfying the second?"
>
> We apologize for the typo and confusion. Please check our revision now where we have rephrased this a bit. It is possible to have mixed skip-connections as the reviewer mentioned, but for simplicity at the moment we just require that all the neurons with skip-connections have the same activation functions which satisfy one of our conditions.
>
> * "The paper does not describe where the assumptions are used...but if you can sketch/mention how the assumptions come into play in the proofs, that will be more helpful in understanding the meaning of the assumptions."
>
> As the reviewer noted, these assumptions are used in the proof of Lemma 3.2, and hence in our main result Theorem 3.3 (though not directly used here). Basically in proving Lemma 3.2, we used our conditions on activation functions to prove that there exists a set of parameters so that the matrix Psi has full rank. Then we use the analytic property of the activation functions together with Lemma A.1 to establish the result on the measure-zero set property. The condition on the training data is used to guarantee that the value of each hidden unit can be chosen to be non-identical for different training samples.
>
> * "Are there any specific reasons for considering cross-entropy loss only? Lemma 3.2 looks general, so this result seems to be applicable to other losses..."
>
> The reviewer is right. Indeed our result holds for other convex loss functions. Please check our extension to this setting in Section C in the appendix. The reason why we presented our main result with cross-entropy loss in the beginning is because we wanted to keep everything simple, and also because this is the loss actually used in practice.
>
> * "...Figure 2 is a bit misleading because there are hidden nodes with skip connections to only one of the output nodes."
>
> Yes, they are connected to all the hidden units. We apologize for the confusion in Figure 2 as we thought it might look a bit too dense. Please check our revision now where we have updated the figure.
>
> * "For the experiments, how did you deal with pooling layers in the VGG and DenseNet architectures? Does max-pooling satisfy the assumptions? Or the experimental setting doesn’t necessarily satisfy the assumptions?"
>
> It depends. In general, max-pooling can be used above all the neurons with skip-connections in the network. However as the main goal of the experiments is to find out the generalization performance of skip-networks, we did not want to include this part in the paper. Nevertheless, we have added Section G in the appendix to treat this question separately.
>
> * Can you show the “improvement” of loss surface by adding skip connections? Maybe coming up with a toy dataset and network WITH bad local valleys will be sufficient, because after adding N skip connections the network will be free of bad local valleys.
>
> Yes. Please check our Section E in the appendix now, where we provide a visual example of the loss landscape of a small network, before and after adding skip-connections. One can easily see that skip-connections to the output help to smooth the loss landscape and get rid of bad local valleys.
>
> * "In the Assumption 3.1.3, the $N$ in $r \neq s \in N$ means $[N]$?"
> Yes. We fixed the typo. Thanks!
>
> * "In the introduction, there is a sentence “potentially has many local minima, even for simple models like deep linear networks (Kawaguchi, 2016),” which is not true...."
>
> The reviewer is right. It's actually an english issue as we meant non-convexity which previously appears before this term. We removed it now in our revision.
>
> * "Assumption 3.1.3 looked a bit confusing to me at first glance. You might want to add some clarification such as “for example, in the fully connected network case, this means that all data points are distinct.”"
>
> Thanks for another helpful comment. We have updated/improved the statement of this condition a bit. In particular, we require now only the distinctness between the input patches at the same location across different training samples. This is just a subtle change and the current proof of Lemma 3.2 is not affected by this modification. We follow your suggestion by adding the following sentence right below Equation (3):
> "The third condition is always satisfied for fully connected networks if the training samples are distinct. For CNNs, this condition means that the corresponding input patches across different training samples are distinct."

---

### Official Review · AnonReviewer2 · 2018-11-03
**a breakthrough paper on the loss landscape of neural networks**

**Rating:** 8
**Confidence:** 4

**Review:**

The paper analyzes the loss landscape of a class of deep neural networks with skip connections added to the output layer. It proves that with the proposed structure of DNN, there are uncountably many solutions with zero training error, and the landscape has no bad local valley or local extrema.

Overall I really enjoy reading the paper.
The assumptions to aid the proof are very natural and much softer than the existing literature. As far as I’m concerned, the setting is very close to real deep neural networks and the paper is a breakthrough in the area. The experiments also consolidate that the theoretical settings are natural and useful, namely, with enough skip connections and specially chosen activation functions.
The presentation of the paper is intuitive and easy to follow. I’ve also checked all the proof and think it’s brilliantly and elegantly written.

My only complaint is about the experiments. As we all know that both VGG and the sigmoid activation are commonly used DL tools, and why do they fail to generalize when used together? Does the network fail to converge or is it overfitting? The authors should try tuning the parameters and present a proper result. With that said, since the paper is more about theoretical findings, this issue doesn’t influence my recommendation to accept the paper.


Minor issues:
I think it’s better to formally define “bad local valley” somewhere in the paper. From what I read, the definition of “bad local valley” is implied by the abstract and in the proof of Theorem 3.3(2), but I did not find a formal definition anywhere else.
In proof number 4 (of Theorem 3.3), the statement should be “any *principle* submatrices of negative semi-definite matrices are also NSD”, and it’s not true otherwise. But this typo doesn’t influence the proof.
Also, it seems the proof of 3 is somewhat redundant, since local minimum is a special case of your “bad local valley”.
It seems the analysis could not possibly be extended to the ReLU activation, since it will break the analytical property of the function. Just out of curiosity, do the authors have some further thoughts on non-differentiable activations?

---

> ### Author Response · Authors · 2018-11-16
> **Response to AnonReviewer2**
>
> Thank you very much for the support. Below are our answers to your comments/questions in the order that they appear.
>
> Regarding the failure of original VGG with sigmoid activation, we have added a discussion on this issue under Section F in the appendix (please see also our response to AnonReviewer3 on the 10% accuracy matter).
> Basically, we have observed that the network in this case converges to a constant zero classifier, regardless of our effort in tuning the learning rate. This behavior is actually not restricted to the specific architecture of VGG, but has been shown before as an issue of sigmoid activation when training plain networks with depth > 5, see e.g. [1].
>
> Answers to minor issues:
> Actually the definition of bad local valleys has previously appeared just above Theorem 3.3 in the text. However we follow the reviewer's suggestion by putting this in a formal definition 3.3 now.
>
> "In proof number 4 (of Theorem 3.3), the statement should be “any *principle* submatrices of negative semi-definite matrices are also NSD”, and it’s not true otherwise. But this typo doesn’t influence the proof."
> Yes, the reviewer is completely right. We fixed this typo. Thanks!
>
> "Also, it seems the proof of 3 is somewhat redundant, since local minimum is a special case of your “bad local valley”."
> We agree. We keep it there as we wanted to make all our statements and results become clear and as rigorous as possible.
>
> "It seems the analysis could not possibly be extended to the ReLU activation, since it will break the analytical property of the function. Just out of curiosity, do the authors have some further thoughts on non-differentiable activations?"
> Thank you for an interesting question. At the moment, we do not really have a clear clue how to extend the result to general non-differentiable activations, so this could be an interesting question for future research.
> For ReLU, we think that it might be possible to exploit the fact that softplus can approximate ReLU arbitrarily well, and so perhaps a limiting argument on their corresponding loss functions can be helpful..
>
> [1] Understanding the difficulty of training deep feedforward neural networks. Xavier Glorot, Yoshua Bengio. ICML 2010.

---

### Official Review · AnonReviewer3 · 2018-11-05
**good progress; but simulation requires some work**

**Rating:** 7
**Confidence:** 4

**Review:**

This paper shows that a class of deep neural networks have no spurious local valleys –--implying no strict local-minima. The family of neural networks studied includes a wide variety of network structure such as (a variant of) DenseNet. Overall, this paper makes some progress, improving previous results on over-parametrized networks.

Pros: The flexibility of the network structure is an interesting point.
Cons: CNN was covered in previous related works (so weight sharing is not a new contribution); DenseNet is not explicitly covered in this work (I mean current DenseNet does not have N skip-connections to output; correct me if wrong).
  The simulation part is not that clear, and I have a few questions that I hope the authors can answer.

Some comments/suggestions:
1) Training error needs to be discussed.
   Page 8 says “This effect can be directly related to our result of Theorem 3.3 that the loss landscape of skip-networks has no bad local valley and thus it is not difficult to reach a solution with zero training error”. This relation is not justified. The implication of Thm 3.3 is that getting zero training error is easier, but the tables are only for test error. Showing training error is the only way to connect to Thm 3.3. I expect to see a high training error for C-10, original VGG and sigmoid activation functions, and zero training error for both skip-SGD (rand) and skip-SGD (SGD).
    This paper has no theory on generalization, thus if a whole section is just about “investigating generalization error”, then the connection to theoretical parts is weak --btw, one connection is the comparison of two algorithms, which fits the context well, and thus interesting (though comparison result itself probably not surprising).

2) Data augmentation.
  “Note that the rand algorithm cannot be used with data augmentation in a straightforward way and thus we skip it for this part.” Why?
   With data augmentation, is M still larger than N? If yes, then the number of added skip connection is different for C-10 and C-10-plus, which is not mentioned in the instruction of Table 2.

3)It may be better to mention explicitly that "it is possible to have bad local min" –perhaps in abstract and/or introduction.
  --Although “no sub-optimal strict local minima” is mentioned, readers, especially non-optimizers, might not notice "strict".
  --In fact, in the 1st round read, I do not have a strong impression of "strict". Later I realized it. Mentioning this can be helpful.

4) Some references I suggest to include:
   [R1] Yu, X. and Chen, G. On the local minima free condition of backpropagation learning. 1995.  --related work.
   [R2] Lu, H., Kawaguchi, K. Depth creates no bad local minima. 2017. --also deep nets.
   [R3] Liang, S., Sun, R., Li, Y., & Srikant, R. "Understanding the loss surface of neural networks for binary classification." 2018. --Also study SoftPlus neurons.
   [R4] Nouiehed, M., & Razaviyayn, M. Learning Deep Models: Critical Points and Local Openness. 2018. --also deep nets.

Minor questions:
  --Exact 10% test accuracy for a few cases. Why exact 10%?

---

> ### Author Response · Authors · 2018-11-16
> **Response to AnonReviewer3**
>
> Thank you for the feedback. Below are answers to your comments/questions by their numbering.
>
> 1) We agree with the reviewer about the training error matter. Thus we have added Section F in the appendix to discuss training error in details. As expected, the training error is zero except the case where sigmoid activation is used with original VGGs from Table 2 or original CNN13 from Table 1.
> Moreover, we show in this section that adding skip-connections to the output is also helpful for training extremely deep (narrow) networks with softplus activation. This together show that skip-connections are helpful for training deep networks with both sigmoid and softplus activation. In Section E in the appendix, we provide a visual example of the loss landscape of a small network, before and after adding skip-connections, where one can see that adding skip-connections to the output layer help to smooth the loss surface and get rid of bad local valleys, which is helpful for local search algorithms like SGD to succeed.
>
> 2) As described in our experiments, the number of skip-connections is fixed to M=N in both cases (with and without data-augmentation), where N is the size of the original data set. We quote the following sentence from our experimental section for the convenience of the reviewer:
> "...we aggregate all neurons of all the hidden layers in a pool and randomly choose from there a subset of N neurons to be connected to the output layer...".
> In the setting of data-augmentation, at each training iteration the network uses additional examples (randomly) generated from the original dataset, and thus it is not clear in this case how the number of training samples should be defined. That's why we fixed the number of skip-connections in both cases to be the size of the original data set.
>
> 3) We agree that this might be overlook by non-optimizers. Nevertheless we want to keep our abstract short and precise. Thus we have added the following sentence in the introduction to make this further clear:
> "We note that this implies the loss landscape has no strict local minima, but theoretically non-strict local minima can still exist."
>
> 4) We have included the references suggested by the reviewer, and can add more detailed comparisons if the reviewer think that it's necessary.
>
> Regarding 10% test accuracy, we added a discussion on this issue under Section F in the appendix. Briefly, the reason, as observed in our experiments, is that the network converges quickly to a constant zero classifier (i.e. the output of last hidden layer converges quickly to zero), and thus the training/test accuracy converge to 10% and the cross-entropy loss in Equation (2) converges to − log(1/10). We realized later that this is actually a known issue of sigmoid activation when training plain networks with depth > 5, as pointed out earlier by Glorot & Bengio [1].
>
> [1] Understanding the difficulty of training deep feedforward neural networks. Xavier Glorot, Yoshua Bengio. ICML 2010.

---

> > ### Comment · AnonReviewer3 · 2018-11-24
> > **some concerns on experiments remain**
> >
> > Overall, I think this paper is quite nontrivial since a rigorous mathematical proof is indeed the interesting part and often quite difficult, and the idea of having flexible skip connections is interesting. But perhaps it is less than a breakthrough due to prior related work on CNN.
> >
> > I'd like to thank the authors for the effort in improving the paper.  My concerns are partially but not fully addressed, as explained below.
> >
> >  1)  As I said, "This paper has no theory on generalization, thus if a whole section is just about test error, then the connection to theoretical parts is weak." It is good that the authors add the training error table Table 4, but Table 4 appears in the appendix. I have to compare Table 2 and Table 4 a few times, when I re-read the paper. Isn't it better to put Table 4 in the main body?  That may be a hard choice, as some parts need to moved into the appendix. But having Table 2 and 4 separately is strange. In fact, from a theoretician's perspective, having solely Table 2 in the main body while having Table 4 in the appendix is fine (though some practitioners don't think so). Anyway, having both may be better.
> >     In addition, "The training error is zero in all cases, except when the original VGG models are used with sigmoid activation function" is inconsistent with Table 4, which shows for SoftPlus the training accuracy is also 10%. After comparing with Table 2, I noticed it is probably due to typo. All SoftPlus results in Table 4 should be 100. These typos probably won't appear if Table 4 is near Table 2.
> >
> > (2) I am not satisfied with this explanation on data augmentation.
> >   First, there are two types of augmentation: "at each training iteration the network uses additional examples" refers to online augmentation; increasing the dataset size and use them in all iterations is off-line augmentation. Clearly, for off-line augmentation, N is increased.
> >   Second, note that for SGD, some statisticians often refer to one-pass over all data, while many optimizers often refer to multi-passes. In other words, for online augmentation, these statisticians would count additional data (even just used once) into N.
> >      It is not clear why the authors need to include the experiments with data augmentation in Table 2. For the purpose of illustrating their point, experiments with data augmentation are not necessary --this is a theory paper after all. From a theory perspective, it may break the assumption. The easiest way to fix is just to remove the columns on data augmentation.  If not,  it requires further explanation such as "yes, it does not satisfy the assumption, but we just want them to be more comprehensive", or "simple data changes do not affect the training much, so it is close to theory". Anyhow, none of them is very satisfying for me.

---

> > > ### Author Response · Authors · 2018-11-24
> > > **Thank you for your feedback**
> > >
> > > We thank the reviewer for the response and further comments on the presentation issue of our experimental results.
> > >
> > > We have updated the paper accordingly by taking into account both comments of the reviewer together. Regarding comment 2), we removed the column of data-augmentation in Table 2, and moved them to the appendix for interested readers. We then used this space to show the training accuracy of all models, which is recommended by the reviewer in comment 1). We hope that this becomes more clear now.
> > >
> > > We thank the reviewer again and we welcome further comments on our paper.

---

### Meta-Review · Area_Chair1 · 2018-12-12

**Confidence:** 4
**Recommendation:** Accept (Poster)

**Metareview:**

This paper introduces a class of deep neural nets that provably have no bad local valleys. By constructing a new class of network this paper avoids having to rely on unrealistic assumptions and manages to provide a relatively concise proof that the network family has no strict local minima. Furthermore, it is demonstrated that this type of network yields reasonable experimental results on some benchmarks. The reviewers identified issues such as missing measurements of the training loss, which is the actual quantity studied in the theoretical results, as well as some issues with the presentation of the results. After revisions the reviewers are satisfied that their comments have been addressed. This paper continues an interesting line of theoretical research and brings it closer to practice and so it should be of interest to the ICLR community.